# Paraventricular hypothalamus mediates diurnal rhythm of metabolism

Eun Ran Kim [1,10], Yuanzhong Xu [1,10], Ryan M. Cassidy [1,2], Yungang Lu [1], Yongjie Yang[3], Jinbin Tian[1,4], De-Pei Li [5], Rachel Van Drunen [1,2], Aleix Ribas-Latre[1], Zhao-Lin Cai [6,7], Mingshan Xue [6,7], Benjamin R. Arenkiel[6,8], Kristin Eckel-Mahan [1], Yong Xu [3] & Qingchun Tong [1,2,9 ✉]

Defective rhythmic metabolism is associated with high-fat high-caloric diet (HFD) feeding, ageing and obesity; however, the neural basis underlying HFD effects on diurnal metabolism remains elusive. Here we show that deletion of BMAL1, a core clock gene, in paraventricular hypothalamic (PVH) neurons reduces diurnal rhythmicity in metabolism, causes obesity and diminishes PVH neuron activation in response to fast-refeeding. Animal models mimicking deficiency in PVH neuron responsiveness, achieved through clamping PVH neuron activity at high or low levels, both show obesity and reduced diurnal rhythmicity in metabolism. Interestingly, the PVH exhibits BMAL1-controlled rhythmic expression of GABA-A receptor γ2 subunit, and dampening rhythmicity of GABAergic input to the PVH reduces diurnal rhythmicity in metabolism and causes obesity. Finally, BMAL1 deletion blunts PVH neuron responses to external stressors, an effect mimicked by HFD feeding. Thus, BMAL1-driven PVH neuron responsiveness in dynamic activity changes involving rhythmic GABAergic neurotransmission mediates diurnal rhythmicity in metabolism and is implicated in diet-induced obesity.

[1] Brown Foundation Institute of Molecular Medicine, University of Texas McGovern Medical School, Houston, TX 77030, USA. [2] Graduate Program in Neuroscience of MD Anderson and UTHealth Graduate School, Houston, TX 77030, USA. [3] Children's Nutrition Research Center, Department of Pediatrics, Baylor College of Medicine, Houston, TX, USA. [4] Department of Integrative Physiology and Pharmacology, University of Texas McGovern Medical School, Houston, TX 77030, USA. [5] Department of Critical Care and Respiratory Care, Division of Anesthesiology, Critical Care and Pain Medicine, University of Texas MD Anderson Cancer Center, Houston, TX 77030, USA. [6] Department of Neuroscience, Baylor College of Medicine, Houston, TX 77030, USA. [7] Cain Foundation Laboratories, Jan and Dan Duncan Neurological Research Institute at Texas Children's Hospital, Houston, TX 77030, USA. [8] Jan and Dan Duncan Neurological Research Institute, Texas Children's Hospital, Houston, TX 77030, USA. [9] Department of Neurobiology and Anatomy of McGovern Medical School, University of Texas Health Science Center at Houston, Houston, TX 77030, USA. [10]These authors contributed equally: Eun Ran Kim, Yuanzhong Xu. ✉email: qingchun.tong@uth.tmc.edu

Through evolutionary adaptation to light–dark cycles, animals have developed diurnal rhythms in metabolism, feeding, and locomotion. As a typical nocturnal species, mice exhibit higher levels of energy expenditure, feeding, and locomotion during periods of dark, compared to light[1]. Similarly, individual cells also exhibit diurnal expression patterns of functionally important genes. One such a group of genes are clock genes, which, through rhythmic regulation of numerous downstream targets, control the cyclical function of cells[2]. Previous studies on circadian biology with a focus on diurnal patterns of locomotion have identified the clock gene expression in the superachiasmatic nucleus (SCN) as a major mechanism underlying light entrainment of diurnal patterns in both locomotion and clock gene expression[3–5]. Notably, mice with dysfunctional clock genes also exhibit defects in diurnal patterns in feeding and metabolism[6–8], suggesting an important role for clock genes in mediating diurnal rhythms in metabolism, in addition to diurnal locomotion. Recent studies suggest that ageing and high-caloric diet (HFD)-induced obesity (DIO) is associated with diminished diurnal rhythms in metabolism and feeding[9,10], and importantly, scheduled feeding, i.e., limiting feeding only within dark periods, greatly improves metabolism despite similar energy intake[11,12]. Together, these findings reveal a critical function for clock gene-related diurnal rhythms in metabolism and health. However, despite extensive studies on circadian biology, the neural basis governing diurnal patterns in metabolism remains elusive.

The hypothalamus is a well-established brain site that regulates energy expenditure and feeding behavior[13]. Discrete groups of neurons in the hypothalamus are capable of sensing changes in various hormones and nutritional status, and function to dynamically adjust food intake and metabolism to maintain energy homeostasis[14]. The paraventricular hypothalamus (PVH) has emerged as a major brain area that regulates both feeding and energy expenditure through integration of inputs from other brain regions[15]. In addition, agouti-related protein (AgRP) neurons in the arcuate nucleus (Arc), and GABAergic neurons in the lateral hypothalamus (LH), promote feeding through GABAergic signaling to the PVH[16–18]. Interestingly, AgRP neuron activity is increased during dark periods in mice[19]. Apart from AgRP neurons, our previous studies have shown that hypothalamic non-POMC, non-AgRP neurons send abundant projections to, and contribute significant GABAergic input to the PVH[20]. Consistently, Arc TH neurons also similarly promote feeding via GABAergic projections to the PVH[21]. Notably, deletion of Arc leptin receptor-expressing neurons, most of which are GABAergic neurons and project to the PVH[22], leads to disruption of diurnal patterns of feeding and locomotion[23]. These observations point to a potential role for GABAergic inputs to the PVH in diurnal feeding and metabolism. However, it remains to be established how GABAergic inputs to the PVH are integrated to control metabolism and feeding behaviors.

PVH neurons are known to express circadian genes including BMAL1 and CLOCK, which can be entrained by feeding[24]. However, the relevance of PVH circadian gene expression to regulating metabolism, and important downstream effectors for these circadian genes, is not clear. Notably, the PVH expresses abundant GABA-A receptor γ2 subunits[20], which represent a major subunit of GABA-A receptors[25]. Previous studies suggest that GABA signaling and the expression of various GABA-A receptor subunits in other brain areas exhibit diurnal patterns[26,27], suggesting a potential role for GABAergic action in mediating diurnal rhythms. Here we show that PVH neurons exhibit BMAL1-controlled diurnal GABA-A receptor γ2 subunit expression. BMAL1 deletion diminished PVH neuron activation in response to fast–refeeding, reduces diurnal metabolic rhythmicity, and causes obesity development. Animal models with

PVH neuron activity chronically clamped at a high or low level, both with abolished PVH neuron activation to fast–refeeding, exhibit reduced diurnal rhythmic metabolism and obesity. Importantly, animals with loss of diurnal γ2 expression also exhibit obesity with reduced diurnal rhythmicity in metabolism and feeding. Further in vivo fiber photometry data show that PVH neurons are rapidly activated by external stressors, which is greatly attenuated by HFD feeding or BMAL1 deletion. These results identify BMAL1-driven PVH neuron responsiveness as a neural basis in regulating diurnal rhythmic metabolism.

## Results

### Impact of PVH BMAL1 deletion on neurons and metabolism.
BMAL1 is abundantly expressed in the PVH and its expression can be entrained by feeding[24]. To examine the function of BMAL1 in the PVH and to avoid a concern from developmental compensation, we aimed to delete its expression in adult mice by stereotaxic delivery of adeno-associated viral vectors expressing Cre (AAV-Cre-GFP) to the PVH of $Bmal1^{flox/flox}$ mice. In controls with bilateral AAV-GFP delivery to the PVH (Fig. 1a), BMAL1 was expressed in the majority of PVH neurons (Fig. 1b, c). In contrast, in AAV-Cre-GFP-injected mice (Fig. 1d), BMAL1 was deleted throughout the PVH (Fig. 1e, f). Compared to controls, deletion of BMAL1 (KO) in the PVH led to a dramatic increase in body weight (Fig. 1g). Six weeks after viral injections, body weight gain increased up to 12 g, showing rapid obesity development, whereas controls gained little body weight (Fig. 1h). To better understand the mechanisms underlying the observed obesity, we measured energy expenditure and feeding at an early time point 2–3 weeks post viral injections with no or little body weight difference between groups. Whereas control mice exhibited a robust diurnal rhythm, i.e. high levels of $O_2$ consumption during night and low during day, KO mice showed dramatic reduction in $O_2$ consumption rhythms (Fig. 1i). The reduced rhythmicity was more evident when the difference in $O_2$ consumption between day and night, which was greater in controls, compared to KOs (Fig. 1j). A similar reduction in feeding rhythm was also observed (Fig. 1k, l), although the difference in day/night food intake between groups was not statistically significant (Fig. 1l). Notably, daily average $O_2$ consumption was lower in KOs (Fig. 1m) but no differences in daily average food intake was observed between groups (Fig. 1n). In addition, diurnal locomotion pattern was also reduced by BMAL1 deletion in the PVH (Supplementary Fig. 1a). Similar changes were also observed when the same measurement was performed 8–9 weeks after viral delivery (Supplementary Fig. 1b–e). These results suggest that deletion of BMAL1 in the PVH disrupts diurnal rhythmicity in metabolism, increases feeding efficiency, and causes obesity.

PVH neurons are known to exhibit diurnal activity patterns and respond to metabolic changes[28]. To examine whether BMAL1 deletion alters PVH neuron function related to metabolism, we measured PVH neuron activity changes in response to fast–refeeding. Consistent with previous results[29], in control mice with bilateral delivery of AAV-GFP vectors to the PVH (Fig. 1o, left panels), PVH neurons exhibited minimal c-Fos expression during fasting conditions but showed abundant c-Fos expression upon refeeding (Fig. 1o, right panels), suggesting PVH neuron activation in response to fast–refeeding. In contrast, PVH neurons with BMAL1 deletion, which was mediated by AAV-Cre-GFP delivery to the PVH (Fig. 1p, left panels), exhibited no increase in c-Fos expression after fast–refeeding (Fig. 1p, right panel, and Fig. 1q). Thus, BMAL1 deletion in PVH neurons leads to loss of responsiveness of these neurons to metabolic changes.

### Clamping PVH neuron activity at lower levels disrupted diurnal patterns in metabolism and caused obesity. The

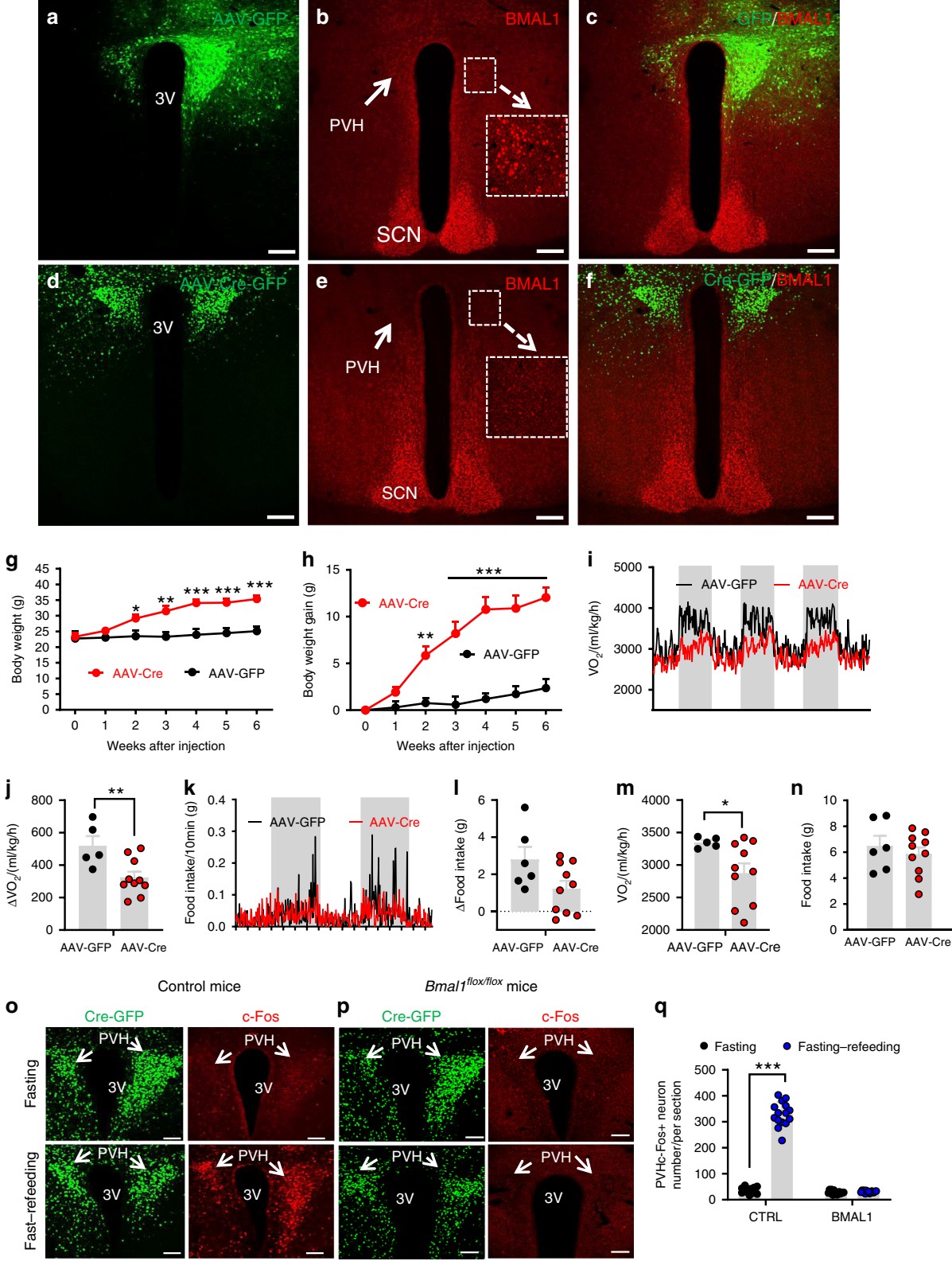

aforementioned data suggest that BMAL1 in the PVH is required for neuron responsiveness, diurnal rhythms in metabolism, and normal body weight regulation. To confirm whether defective PVH neuron responsiveness has a causal role in loss of diurnal rhythms and obesity, we aimed to generate mouse models with similar loss of PVH neuron response to metabolic changes. We reason that one effective way is to "lock" the activity of PVH neurons at a constant level, preventing changes in their activity to metabolic changes. Toward this, we expressed the inward rectifying Kir2.1 channel, a potassium channel known to reduce neuron activity[30], in the PVH (Fig. 2a). We bilaterally delivered either conditional AAV-FLEX-mCherry vectors as controls (data not shown) or conditional AAV-EF1a-DIO-Kir2.1-P2A-dTomato vectors (Fig. 2b, left panels) to the PVH of *Sim1-Cre* mice.

**Fig. 1 Adult deletion of PVH BMAL1 disrupted diurnal metabolism and caused obesity. a–n** $Bmal1^{flox/flox}$ mice (8–10 weeks old) received bilateral injections of AAV-GFP or AAV-Cre-GFP and were used for immunostaining BMAL1 expression and body weight studies. **a–c** Brain sections from AAV-GFP-injected mice were examined for GFP expression (**a**), BMAL1 (**b**), and merged (**c**). **d–f** Brain section from AAV-Cre-GFP-injected mice were examined for GFP (**d**), BMAL1 (**e**), and their merged expression (**f**). Arrows pointing to BMAL1 expression in GFP (**b**) and Cre-injected mice (**e**). Insets in **b** and **e** showing BMAL1 expression in a higher magnification. PVH paraventricular hypothalamus, 3V the third ventricle, SCN superachiasmatic nucleus. Scale bar = 200 μM. **g–h** Weekly body weight (**g**, two-way ANOVA, $n = 5$ for GFP and 10 for Cre, $F(6, 84) = 3.382$, ***$p < 0.0001$, body weight at 6 weeks between GFP vs Cre) and net increases in body weight (**h**, two-way ANOVA, $n = 5$ for GFP and 10 for Cre, $F(6, 84) = 7.051$, ***$p < 0.0001$, body weight at 6 weeks between GFP vs Cre) were followed. **i–n** Real time traces showing energy expenditure (**i**) and feeding (**k**) measured by CLAMS 3 weeks after viral delivery, comparison in difference between day and night periods of energy expenditure (**j**, unpaired two-tailed Student's $t$-test, $n = 5$ for GFP and 10 for Cre, $t = 3.014$, d.f. = 13, $p = 0.01$) and feeding (**l**), and comparison in daily feeding (**m**, unpaired two-tailed Student's $t$-test, $n = 5$ for GFP and 10 for Cre, $t = 2.168$, d.f. = 13, *$p = 0.0493$) and energy expenditure (**n**) between groups. **o–q** Control or $Bmal^{flox/flox}$ mice with AAV-Cre-GFP injections were either overnight fasted alone or overnight fasted with 2 h refeeding, and then immunostained for c-Fos. Representative expression of GFP and c-Fos in the PVH in GFP (**o**) and BMAL1 deleted mice (**p**). At least three mice with five sections each containing the PVH were used for counting the number of c-Fos in the PVH. Arrows point to the PVH proper; 3V the third ventricle. **q** Comparison in average number of c-Fos-positive neurons in the PVH (two-way ANOVA, $n = 15/$each, $F(1, 56) = 536.8$, ***$p < 0.0001$ for GFP fasting vs fasting–refeeding, $p = 0.9918$ for Cre fasting vs fasting–refeeding). All data presented at mean ± SEM. Source data are provided as a Source Data file.

Compared to controls, Kir2.1 expression reduced resting membrane potential (Supplementary Fig. 2a–c), input resistance (Supplementary Fig. 2d–f), and significantly increased the size of minimum currents required to be injected to elicit action potential, i.e., rheobase (Supplementary Fig. 2g–i). These data demonstrate that expression of Kir2.1 effectively reduces the PVH neuron activity. Given that the recordings were performed 4–6 weeks post viral injection, these data suggest that Kir2.1 expression chronically reduces PVH neuron activity.

Kir2.1 expression in the PVH abolished c-Fos induction during fast–refeeding (Fig. 2b, right panels), which is in stark contrast to induction of abundant c-Fos expression in the PVH after fast–refeeding (Fig. 2c, d), confirming that Kir2.1 expression leads to loss of PVH neuron response to metabolic changes. Kir2.1 expression led to a dramatic increase in body weight, compared to controls (Fig. 2e) and rapid body weight gain (Fig. 2f), suggesting obesity development. We also performed measurements of energy expenditure (Fig. 2g–i) and food intake (Fig. 2j–k) at 2–3 weeks (Fig. 2g, i) and 8–9 weeks (Fig. 2h, i) time points after viral delivery. Measurement at an earlier time point with no or little body weight difference is to achieve a more effective comparison in $O_2$ consumption. Expression of Kir2.1 led to a dramatic reduction in diurnal rhythm of $O_2$ consumption (Fig. 2g–h), and the difference in $O_2$ consumption between day and night periods was smaller in Kir2.1 mice compared to controls (Fig. 2i) at both time points although not reaching statistical significance at 2–3 week time point. The diurnal rhythm in food intake was also reduced (Fig. 2j), although not significant (Fig. 2k), likely due to large variation in feeding behavior. Of note, at 2–3 weeks post viral delivery, average $O_2$ consumption was significantly reduced in Kir2.1 mice (Fig. 2l) whereas no difference in feeding was observed (Fig. 2m), suggesting an increased feeding efficiency as the major drive for the obesity development. In addition, diurnal rhythmicity in locomotion was also diminished by Kir2.1 expression in the PVH (Supplementary Fig. 2j). Collectively, these data demonstrate that clamping PVH neuron activity at a low level disrupts PVH neuron responsiveness, reduces diurnal rhythmicity in metabolism, leading to obesity development.

**Clamping PVH neuron activity at higher levels on metabolism.** Our data from the Kir2.1 mouse model show that chronic inhibition of PVH neuron activity reduces neuron responsiveness, disrupts diurnal rhythms in metabolism, and causes obesity. Since PVH neuron inhibition is predicted to cause obesity, these data cannot rule out the possibility that the observed obesity is simply an outcome of neuron inhibition. To verify that Kir2.1-induced obesity is caused by defective neuron responsiveness, we next aimed to examine whether clamping PVH neuron activity at higher levels also disrupts neuron responsiveness, reduces diurnal rhythms in metabolism, and promotes obesity. Towards this, we bilaterally delivered conditional AAV vectors encoding a slow inactivation and low-threshold sodium channel from bacteria (NachBach)[31,32] to the PVH of Sim1-Cre mice. In PVH Sim1 neurons, we verified cell type-specific expression of Cre-dependent AAV-EF1a-FLEX-EGFP-P2A-mNachBac (Fig. 3a, left panels). Electrophysiological recordings of PVH neurons showed, as expected, typical action potentials in control Sim1 neurons (Supplementary Fig. 3a). Sim1 neurons with NachBac expression showed action potentials with lower threshold and much slower inactivation kinetics, resulting in expanded action potential widths (Supplementary Fig. 3b), consistent with previous reports[30,31]. Further recordings showed that Sim1 neurons expressing NachBac exhibited a significantly lower threshold for action potential firing (Supplementary Fig. 3c) and more spontaneous firing (Supplementary Fig. 3d). Since these recordings were performed 4–6 weeks after viral delivery, we concluded that targeted NachBac expression chronically clamps PVH neurons at higher levels.

Consistent with its function in increasing neuron activity, NachBac expression in the PVH dramatically increased c-Fos expression, regardless of fast or fast–refeeding conditions (Fig. 3a, right panels), which is in contrast to controls with a clear increase of c-Fos expression in fast–refeeding, compared to minimal c-Fos expression in the fasting condition (Fig. 3b, c). These results confirm that NachBac expression chronically increases neuron activity level and leads to loss of PVH neuron response to metabolic changes.

Given the known role of PVH neurons in feeding inhibition, we expect that chronic activation of these neurons may reduce body weight. However, similar to the Kir2.1 mice with lowered PVH neuron activity and obesity development, NachBac mice gained significantly more body weight compared to controls (Fig. 3d) and by 8 weeks after AAV delivery, these mice had a net weight gain of nearly 15 g relative to controls (Fig. 3e), suggesting rapid obesity development. To effectively compare $O_2$ consumption, we subjected these mice to Comprehensive Lab Animal Monitoring System (CLAMS) measurements during week 3 after viral delivery with slight body weight difference between groups. In contrast to a strong diurnal $O_2$ consumption pattern observed in controls, NachBac mice exhibited reduced diurnal rhythm (Fig. 3f). This reduced rhythmicity was much more evident when the difference in day and night $O_2$ consumption was compared between controls and the NachBac group (Fig. 3g). Specifically, a

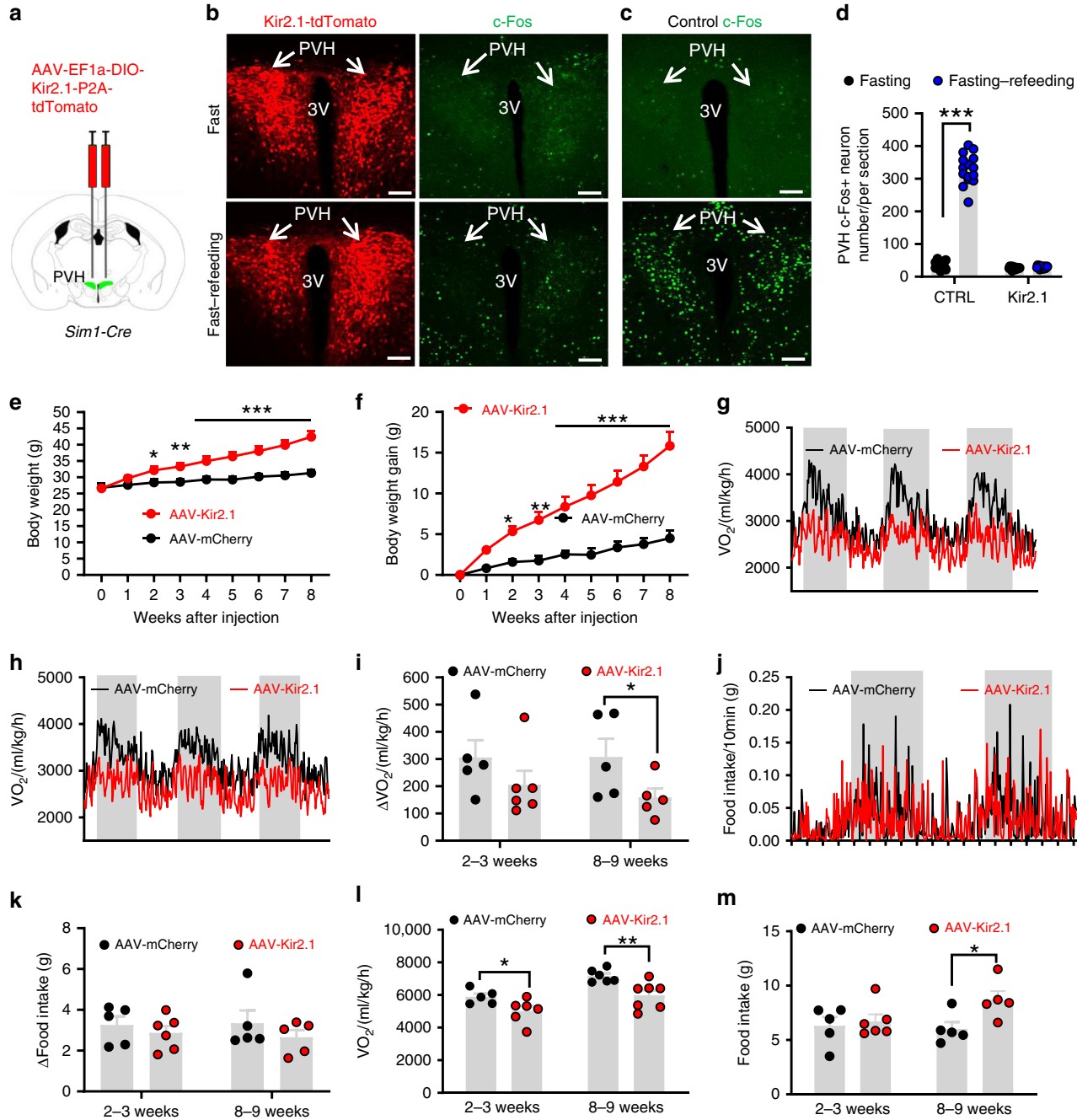

**Fig. 2 Clamping PVH neuron activity at a low lever disrupted diurnal metabolism and caused obesity.** *Sim1-Cre* mice (8–10 weeks old) received injections of AAV-FLEX-mCherry or AAV-DIO-EF1a-Kir2.1-P2A-dTomato vectors to bilateral PVH and used for studies. **a** Diagram showing injections of viral vectors to the PVH of *Sim1-Cre* mice. **b**–**d** Expression of dTomato in the PVH after the Kir2.1 virus expression (**b**, left panels), and representative expression of c-Fos expression in the PVH after overnight fasting alone (top panels) or overnight fast with 2 h refeeding (bottom panels) in Kir2.1 mice (**b**, right panels) and control (**c**). At least three mice with five sections containing the PVH were used for counting the number of c-Fos in the PVH. Arrows point to the PVH proper. **d** Comparison of average number of c-Fos positive neurons in the PVH (two-way ANOVA, $n = 15$ each, $F_{(1, 56)} = 544.4$, ***$p < 0.0001$, control fasting vs fasting–refeeding; and $p = 0.9909$ Kir2.1 fasting vs fasting–refeeding). **e**–**m** Weekly body weight (**e**, two-way ANOVA, $n = 5$ for control and 6 for Kir2.1, $F_{(8, 171)} = 4.022$, ***$p < 0.0001$, body weight at 8-9 weeks after injection between Control and Kir2.1) and weekly net body gain (**f**, two-way ANOVA, $n = 5$ for control and 6 for Kir2.1, $F_{(8, 171)} = 6.401$, ***$p < 0.0001$, body weight at 8-9 weeks after injection between Control and Kir2.1) were followed, and real time traces showing energy expenditure measurements at 2–3 weeks (**g**) and 8–9 weeks (**h**), and feeding 2–3 weeks (**j**) after viral delivery; comparison in the difference of energy expenditure (**i**, two-way ANOVA, $n = 5$ for Control and 6 for Kir2.1, $F_{(1, 17)} = 1.138$, *$p = 0.0378$, 8-9 weeks after injection between Control and Kir2.1) and feeding (**k**) between day and night both 2–3 and 8–9 weeks after viral delivery; daily energy expenditure (**l**, two-way ANOVA, $n = 5$ for control and 6 for Kir2.1, $F_{(1, 20)} = 25.99$, *$p = 0.026$ 2-3 weeks after injection and **$p = 0.0059$ for 8-9 weeks after injection between Control and Kir2.1) and feeding (**m**, two-way ANOVA, $n = 5$ for control and 6 for Kir2.1, $F_{(1, 20)} = 3.292$, *$p < 0.0399$, 8-9 weeks after injection control vs Kir2.1) both 2–3 and 8–9 weeks after viral delivery were shown. All data presented at mean ± SEM. PVH paraventricular hypothalamus, 3V the third ventricle. Scale bar = 200 μM. Source data are provided as a Source Data file.

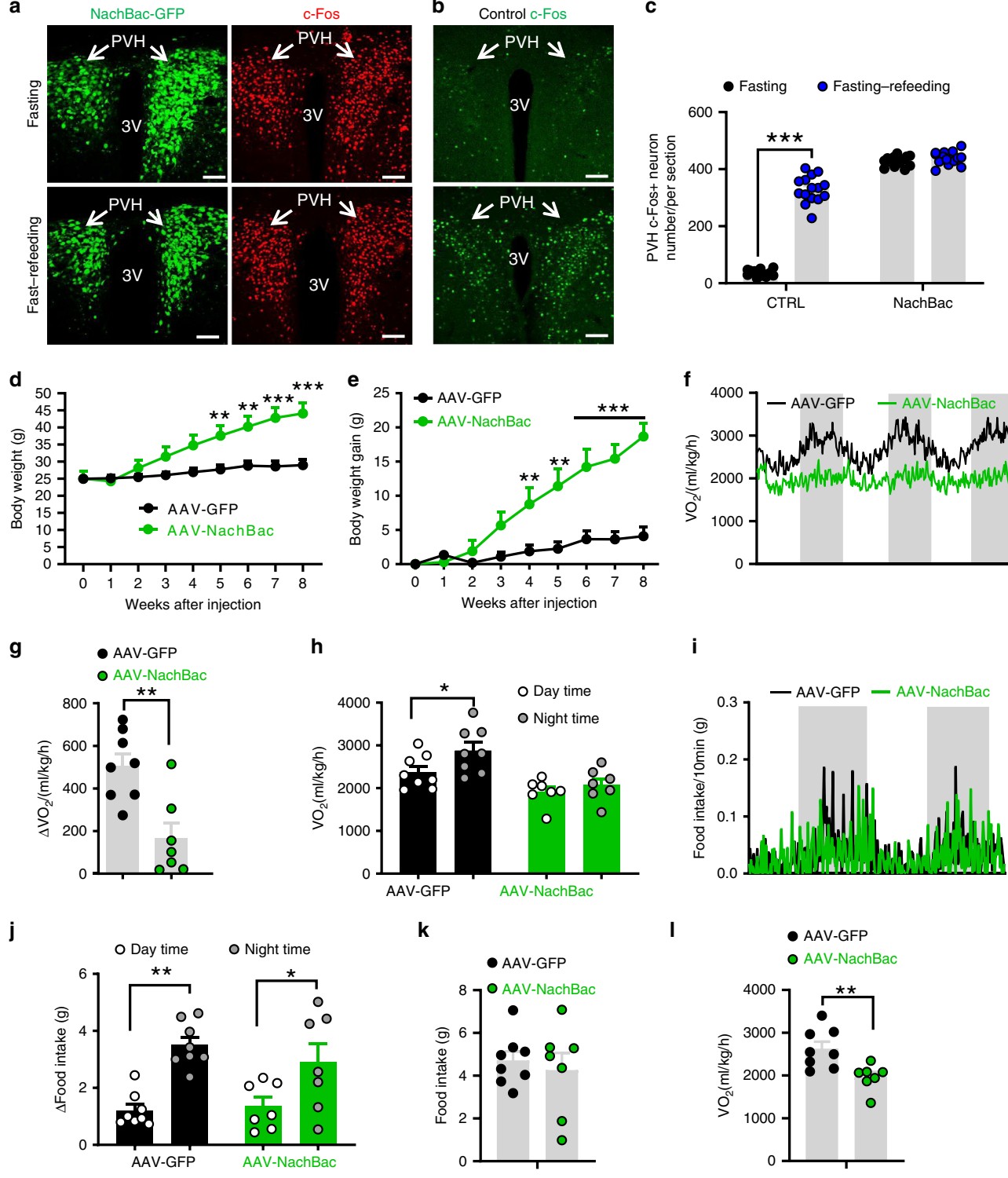

significant more $O_2$ consumption during night than day was observed in controls but no difference was detected in the NachBac group (Fig. 3h). A similar reduction in diurnal feeding rhythm was also observed (Fig. 3i) and the control group consumed greater amount of food during dark periods compared to light periods, while no significant differences were observed in NachBac mice (Fig. 3j). Notably, average daily feeding was not increased in these mice (Fig. 3k), but energy expenditure assessed by $O_2$ consumption was significantly reduced (Fig. 3l). Thus, obesity development in mice with chronic activation of PVH

neurons was caused by reduced energy expenditure. In addition, locomotor activity measured by wheel running in home cages showed that NachBac expression in the PVH reduced diurnal rhythmicity in locomotion (Supplementary Fig. 3e). Given the surprising body weight increasing effects of NachBac, we examined the possibility that NachBac causes overexcitation-induced apoptosis. The expression of cleaved caspase-3, a marker of apoptosis, was not different between NachBac and control groups (Supplementary Fig. 4), arguing against the possibility. These data demonstrate that, similar to Kir2.1-mediated chronic

**Fig. 3 Clamping PVH neuron activity at a high lever disrupted diurnal metabolism and caused obesity.** *Sim1-Cre* mice (8–10 weeks old) received injections of AAV-FLEX-GFP or AAV-EF1a-FLEX-EGFP-P2A-mNachBac vectors to bilateral PVH and used for studies. **a**, **b** Expression of EGFP in the PVH after the NachBac virus expression (**a**, left panels), and representative expression of c-Fos expression in the PVH after overnight fast alone (top panels) or overnight fast with 2 h refeeding (bottom panels) in NachBac mice (**a**, right panels) and control (**b**). **c** Comparison of number of neurons with c-Fos expression in the PVH between fast and fast–refeeding in control and NachBac mice (two-way ANOVA, $n = 15$/each, $F_{(1, 56)} = 385.3$, ***$p < 0.0001$ Control fasting vs refeeding and $p = 0.7212$ NachBac fasting vs refeeding). At least three mice with five sections containing the PVH were used for counting the number of c-Fos in the PVH. Arrows point to the PVH proper. **d–h** Weekly body weight (**d**, two-way ANOVA, $n = 8$ for GFP and 7 for NachBac, $F_{(8, 117)} = 4.252$, ***$p < 0.0001$, body weight at 8 weeks after injection between Control and NachBac), weekly net body gain (**e**, two-way ANOVA, $n = 8$ for GFP and 7 for NachBac, $F_{(8, 171)} = 7.256$, ***$p < 0.0001$, body weight at 8 weeks after injection between Control and NachBac) were followed, real time traces showing energy expenditure (**f**), comparison in the difference of energy expenditure between day and night (**g**, unpaired two-tailed Student's $t$-test, $n = 8$ for GFP and 7 for NachBac, $t = 3.836$, d.f. = 13, **$p = 0.0021$), and differences between day and night periods within groups (**h**, two-way ANOVA, $n = 8$ for GFP and 7 for NachBac, $F_{(3, 26)} = 9.314$, *$p = 0.0488$ control GFP day vs night and $p = 0.8657$, NachBac day vs night). **i**, **j** Real time traces showing feeding (**i**) and comparison in the difference of feeding between day and night (**j**, two-way ANOVA, $n = 8$ for GFP and 7 for NachBac, $F_{(3, 26)} = 9.257$, ***$p = 0.0008$ Control GFP day vs night, and *$p = 0.0458$ NachBac day vs night). **j–l** Daily feeding (**k**) and energy expenditure (**l**, unpaired two-tailed Student's $t$-test, $n = 8$ for GFP and 7 for NachBac, $t = 3.240$, d.f. = 13, **$p = 0.0064$) were compared between groups. PVH paraventricular hypothalamus, 3V the third ventricle. Scale bar = 200 µM. All data presented as mean ± SEM. Source data are provided as a Source Data file.

inhibition of PVH neurons, chronic elevation of PVH neuron activity also disrupts neuron responsiveness, blunts diurnal rhythmicity in metabolism, and causes obesity. Our data from Kir2.1 and NachBac mouse models collectively reveal that the responsiveness, but not absolute activity level, of PVH neurons determines rhythmicity in metabolism and obesity development.

**BMAL1 in diurnal expression of PVH GABA-A γ2 subunit.** Having confirmed that BMAL1 is required for PVH neuron responsiveness and for normal diurnal rhythms in metabolism and body weight regulation, we next aimed to identify important downstream mediators for BMAL1 within the PVH. Given the perceived diurnal patterns of GABA signaling to the PVH[19], we first sought to evaluate the expression pattern of the GABA receptor throughout the day. Indeed, immunostaining of brain sections revealed that the expression level of γ2 subunit was particularly high in the PVH, compared to surrounding regions (Fig. 4a). Results from brains harvested at four different time points (ZT0, 6, 12, and 18) showed that the γ2 expression level was low during the day (ZT6), but high at night (ZT18) (Fig. 4a, b) in the PVH, suggesting that the GABA-A receptor γ2 subunit undergoes a diurnal change in its expression level. To evaluate synaptic activity to PVH neurons, we monitored postsynaptic currents (PSC) for both glutamatergic excitatory EPSCs and GABAergic inhibitory IPSCs. Given the large number of neurons that comprise the PVH, and to ensure that the recorded currents were comparable between mice, we focused on recordings from neurons located within the dorsomedial region of the PVH, since this domain has been shown to receive numerous AgRP and LH neuron inputs[16,17]. Interestingly, using a previously established method[33], we found that while EPSCs showed no difference (Supplementary Fig. 5a, b, Fig. 4c, d), both IPSC amplitude (Fig. 4c) and frequency (Fig. 4d) were significantly increased in night, compared to day. Thus, there is a diurnal increase in the activity of GABAergic input to the PVH during night periods.

Since BMAL1 is known to be a master regulator for circadian gene regulation, we explored whether γ2 is directly regulated by BMAL1 and found two putative E-box-binding sites for BMAL1 in the γ2 promoter region (Fig. 4e, top). We performed ChIP assays and found the evidence that BMAL1 binds to one of the E-boxes (arrow, Fig. 4e, bottom), suggesting that BMAL1 can be recruited to the γ2 promoter region. We further performed luciferase assays and found that BMAL1 and CLOCK increased the γ2 promoter activity both alone and synergistically when both were present (Fig. 4f). To further investigate whether γ2 expression is regulated by BMAL1 in vivo, we compared γ2

expression between day and night in PVH neurons with BMAL1 deletion. While γ2 expression was found to be higher at night (ZT16) compared to day (ZT4) in controls (Fig. 4g, h), no difference in the expression of γ2 was found between day and night with BMAL1 deletion in the PVH (Fig. 4g, h). These results suggest that diurnal expression pattern of γ2 is orchestrated by diurnal expression of BMAL1.

**Impact of adult PVH deletion of GABA-A γ2 subunit.** To examine whether diurnal patterns of γ2 expression mediate the role of BMAL1 in the PVH, we first took a conditional loss-of-function approach and deleted γ2 subunit in the PVH. For this, we bilaterally delivered AAV-Cre-GFP to the PVH of adult (8–10 weeks of age) γ2$^{flox/flox}$ mice (Supplementary Fig. 6a). Brain sections obtained 4 weeks after viral delivery showed that controls with AAV-GFP delivery exhibited abundant γ2 expression in the PVH, but mice that received AAV-Cre-GFP had complete loss of γ2 expression (Supplementary Fig. 6b). To examine the functional consequence of γ2 subunit deletion, we monitored PSCs in the PVH. While we did not detect any significant changes in EPSCs, both amplitudes (Supplementary Fig. 6c) and frequencies (Supplementary Fig. 6d) of IPSCs were greatly reduced in PVH neurons with γ2 deletion, suggesting a specific disruption of GABAergic input by deletion of γ2 subunit. As expected, we also observed that resting membrane potentials were significantly increased following γ2 deletion (Supplementary Fig. 6e), suggesting elevated neuronal excitation presumably due to disrupted inhibitory inputs. Neuron overexcitation has been reported to cause excitotoxicity[34]. So we examined brain sections 12 weeks after AAV-Cre-GFP delivery, and observed signs of cellular damage and tissue loss in the PVH (Supplementary Fig. 6f), indeed suggesting neuronal death. To further explore whether apoptosis contributes to the neuronal death, we immunostained brain sections for cleaved caspase-3 8 weeks after AAV-Cre-GFP delivery. While control sections showed no caspase-3 expression, experimental tissues exhibited abundant caspase-3 expression in the PVH (Supplementary Fig. 6g, arrows). Taken together, these results suggest that deletion of γ2 in the adult PVH caused neuronal apoptosis, likely induced by excitotoxicity.

To examine physiological effects of adult γ2 deletion in the PVH, we next monitored feeding and related behaviors. While control mice exhibited normal diurnal patterns of feeding (Supplementary Fig. 7a) and energy expenditure (Supplementary Fig. 7d), mice with γ2 deletion lost diurnal rhythms in both (Supplementary Fig. 7b, e). Quantitatively, while control mice showed increased levels of feeding (Supplementary Fig. 7c) and

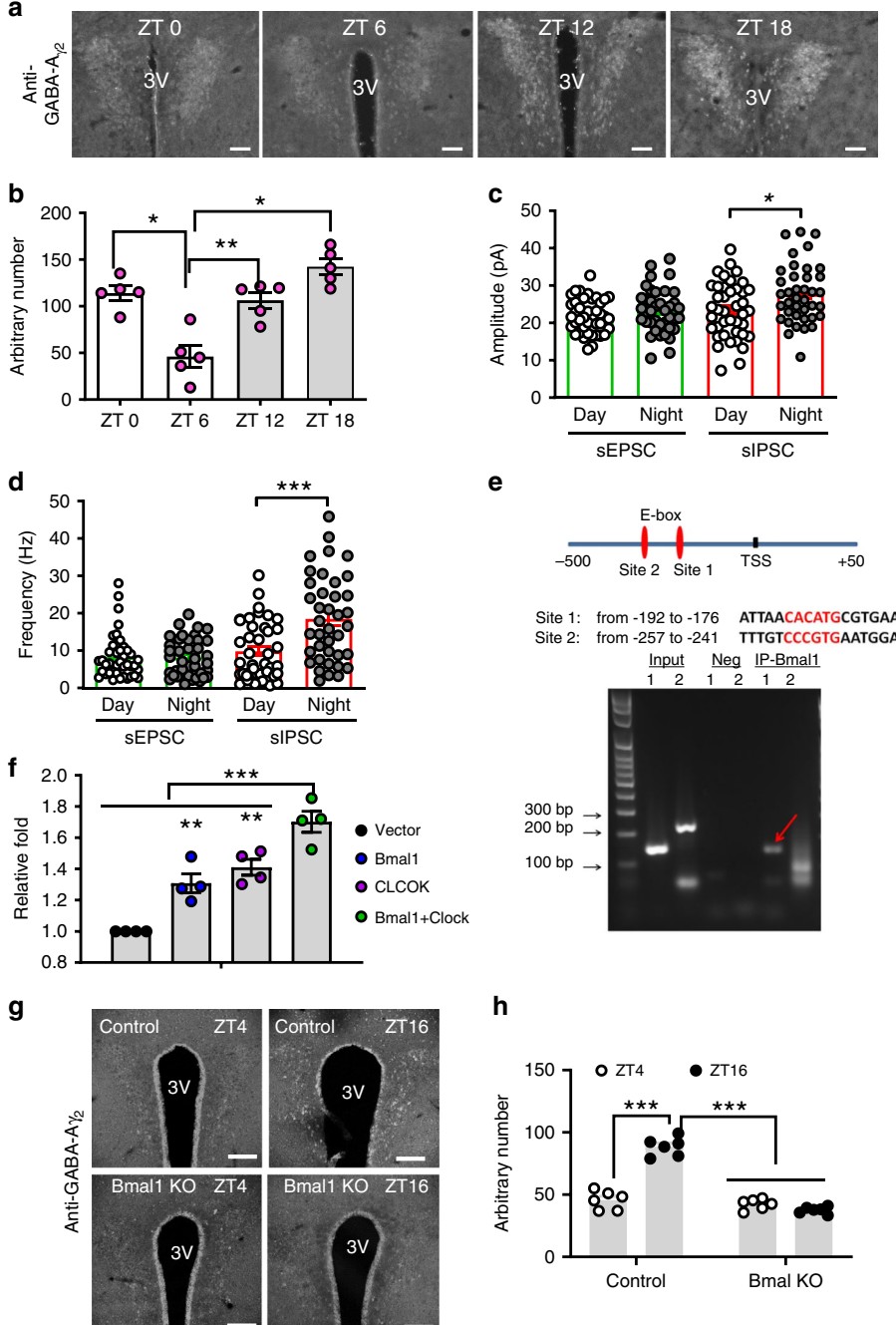

**Fig. 4 Diurnal expression and function of PVH GABA-A γ2 subunit in the PVH. a** Immunostaining of PVH GABA-A γ2 subunit expression at ZT0, 6, 12, and 18 in wild type mice (8–10 weeks old) and (**b**) the quantification data (one-way ANOVA, $n = 5$ each, $F(4, 12) = 0.2464$, *$p = 0.0458$ ZT0 vs ZT6, **$p = 0.0026$ ZT6 vs ZT12, *$p = 0.0258$ ZT6 vs ZT18), showing diurnal expression pattern of γ2 subunit. Quantification of the amplitude (**c**, $n = 41$ for Day and 39 for Night, unpaired two-tailed Student's $t$-test, $t = 2.553$, d.f. $= 78$, *$p = 0.0126$) and frequency (**d**, $n = 41$ for Day and 39 for Night, unpaired two-tailed Student's $t$-test, $t = 3.969$, d.f. $= 78$, ***$p = 0.0002$) of sIPSCs and sEPSCs recorded from PVH neurons during day and night periods (4–6 weeks old). Data are represented as mean ± SEM. Scale bars: 100 μm. PVH paraventricular hypothalamus, 3V the third ventricle, ZT zeitgeber time. **e** Identification of two potential BMAL1-binding E-box motifs in the promoter region of the GABA-A receptor γ2 gene (top) and results of the ChIP analysis showing specific binding of BMAL1 with the site 1 E-Box (arrow). **f** Results from a luciferase assay showing BMAL1 and CLOCK increased the γ2 promoter activity alone and synergistically (one-way ANOVA, $n = 4$ each, $F(3, 12) = 30.93$, **$p = 0.006$ Vector vs BMAL1, **$p = 0.0006$ Vector vs CLOCK, ***$p < 0.0001$ Vector vs BMAL1/CLOCK). **g** Representative γ2 immunostaining from matched PVH sections harvested at day (ZT4) and night (ZT16) in control and BMAL1 deletion mice with AAV-Cre-GFP delivery to the bilateral PVH. **h** Comparison of average γ2 expression in the PVH between day and night in controls and the BMAL1 deleted group (two-way ANOVA, $n = 6$ each, $F(1.963, 9.814) = 95.53$, ***$p < 0.0001$ Control ZT4 vs ZT16, $p = 0.2775$, BMAL1 ZT4 vs ZT16). All data presented at mean ± SEM. Source data are provided as a Source Data file.

energy expenditure (Supplementary Fig. 7f) during dark periods, mice with γ2 deletion exhibited no differences in these parameters between day and night (Supplementary Fig. 7c, f). Associated with loss of diurnal rhythmicity in energy expenditure and feeding, these mice also lost diurnal rhythmicity in locomotion measured by CLAMS (Supplementary Fig. 7g). Supporting this, home cage running wheel activity also lacked a diurnal rhythm in γ2 deletion mice (Supplementary Fig. 7j), compared to controls (Supplementary Fig. 7h) or the body weight-matched HFD-fed group (Supplementary Fig. 7i). Consequently, mice with γ2 deletion developed obesity (Supplementary Fig. 7k). Collectively, these data reveal an essential role for γ2 expression in preventing neuronal death from excitotoxicity and confirm an essential role for PVH neurons in diurnal pattern generation of food intake and energy expenditure.

**Generation of mice with loss of diurnal rhythms of PVH γ2 expression**. Given the issue of PVH neuronal death in mice with γ2 deletion, these mice, despite providing evidence for PVH function in diurnal rhythms in metabolism and obesity, are not suitable to investigate whether γ2 diurnal expression mediates the role of BMAL1. Therefore, we next aimed to generate an animal model with γ2 expression clamped at a constant level (Fig. 5a). Towards this, we generated a new strain of Rosa26-LSL-γ2 knockin mice, in which the wild-type (WT) γ2 cDNA was cloned into the endogenous Rosa26 locus downstream of a lox-transcription blocker (TB)-lox (LSL) cassette (γ2-TB$^{flox/flox}$ mice). This strain allows Cre-dependent expression of WT γ2 with the expression level controlled by the endogenous Rosa26 promoter. The γ2$^{flox/flox}$ mouse strain used for conditional PVH γ2 deletion was previously described[25]. This floxed γ2 allele also contains a point mutation (I77 → F77), which shows no response to the GABA-potentiating drug Zolpidem (Zol), but normal response to GABA. As such, γ2$^{flox/flox}$ mice behave normally, except that they lack response to the sedative effects of Zol, as demonstrated previously[25].

To verify the new γ2-TB$^{flox/flox}$ allele, we generated Sim1-Cre::γ2$^{flox/flox}$::γ2-TB$^{flox/flox}$ mice (Supplementary Fig. 8a). While Sim1-Cre::γ2$^{flox/flox}$ mice showed no γ2 expression in the PVH, compared to controls (γ2$^{flox/flox}$::γ2-TB$^{flox/flox}$ mice), Sim1-Cre::γ2$^{flox/flox}$::γ2-TB$^{flox/flox}$ mice exhibited abundant expression of γ2 (Supplementary Fig. 8b) in the PVH, suggesting effective Sim1-Cre-mediated expression of γ2 driven by the Rosa26 promoter. To examine whether WT γ2 expression driven by the Rosa26 promoter was functional, we performed electrophysiological recordings from PVH neurons. To visually identify Sim1-Cre neurons for recording, we bred these mice to the Ai9 reporter for Cre-dependent expression of tdTomato (i.e. RFP). As expected, in Sim1-Cre mice, evoked IPSCs (eIPSCs) on Sim1 neurons (RFP+) were potentiated by Zol, which was reversible (Supplementary Fig. 8c). However, in Sim1-Cre::γ2$^{flox/flox}$ mice, Sim1 neurons showed greatly reduced eIPSCs, and non-Sim1 neurons (RFP−) exhibited no potentiation in eIPSCs by Zolpidem (Supplementary Fig. 8d), consistent with the effect of I77 → F77 point mutation in the γ2$^{flox/flox}$ allele[25]. Interestingly, in Sim1-Cre::γ2$^{flox/flox}$::γ2-TB$^{flox/flox}$ mice, Sim1 neurons exhibited strong potentiating effects by Zol, while non-Sim1 neurons showed no responses (Supplementary Fig. 8e). These results suggest that Sim1-Cre-mediated expression of WT γ2 subunit driven by the Rosa26 promoter restores responses to Zol in PVH neurons.

To examine patterns of γ2 expression driven by the Rosa26 promoter, we performed immunostaining for γ2 in the PVH of Sim1-Cre::γ2$^{flox/flox}$::γ2-TB$^{flox/flox}$ mice. Unlike the endogenous pattern of diurnal expression normally seen in controls (Fig. 4), levels of γ2 expression were comparable among all four time

points examined (ZT0, 6, 12, and 18) (Supplementary Fig. 8f, g), confirming a non-rhythmic nature of the Rosa26 promoter activity. Interestingly, Sim1-Cre::γ2$^{flox/flox}$ mice were previously shown to have reduced body weight[20], which is in contrast to the obesity phenotype in mice with inducible γ2 deletion by AAV-Cre-GFP shown in this study, suggesting strong compensatory effects from development in Sim1-Cre::γ2$^{flox/flox}$ mice. However, Sim1-Cre::γ2$^{flox/flox}$::γ2-TB$^{flox/flox}$ mice exhibited no alterations in body weight compared to littermate controls (Supplementary Fig. 8h), suggesting functional rescue of γ2 subunit. Surprisingly, Sim1-Cre::γ2$^{flox/flox}$::γ2-TB$^{flox/flox}$ mice showed normal diurnal patterns of feeding and energy expenditure (Supplementary Fig. 8i, j), suggesting that alterations in diurnal γ2 expression pattern from an early neonatal stage fail to alter diurnal regulation in metabolism or body weight.

To eliminate potential developmental compensation, we next disrupted diurnal γ2 expression in the PVH via bilateral delivery of AAV-Cre-GFP to the PVH of 8–10-week-old γ2$^{flox/flox}$::γ2-TB$^{flox/flox}$ mice, thereby replacing the endogenous γ2 expression with Rosa26 promoter-driven γ2 in adult mice (Fig. 5a). As expected, brain sections immunostained for γ2 expression 4 weeks after AAV-Cre-GFP vector delivery showed no γ2 expression in the PVH (Fig. 5b, middle panels), compared to control AAV-GFP-injected mice (Fig. 5b, left panels), whereas AAV-Cre-GFP-injected γ2$^{flox/flox}$::γ2-TB$^{flox/flox}$ mice showed restoration of γ2 expression in the PVH (Fig. 5b, right panels).

Given the excitotoxicity-induced neuronal death observed with γ2 deletion in adults, we next investigated whether γ2 replacement in adult PVH neurons counteracted this effect. For this, we examined cleaved caspase-3 expression 14–15 weeks after AAV-Cre-GFP delivery, a time point much later than the one with neuronal death by adult γ2 deletion shown in Supplementary Fig. 5. In both control and AAV-Cre-GFP-injected mice, we found no evidence of cleaved caspase-3 expression in the PVH (Supplementary Fig. 9), suggesting that ROSA26-driven γ2 expression rescued the observed excitotoxicity-induced apoptosis induced by adult γ2 deletion.

To examine the effect of γ2 replacement on diurnal patterns of GABAergic input to the PVH, we repeated the PSC recording shown in Fig. 4. In contrast to controls, the eIPSC amplitude was not different between day and night periods in AAV-Cre-GFP-injected γ2$^{flox/flox}$::γ2-TB$^{flox/flox}$ mice, consistent with the effect of disrupted diurnal patterns of γ2 subunit expression (Supplementary Fig. 5c, d, Fig. 5c). Surprisingly, the increased eIPSC frequency in controls was also lost in AAV-Cre-GFP-injected γ2$^{flox/flox}$::γ2-TB$^{flox/flox}$ mice (Supplementary Fig. 5c, d, Fig. 5d). Collectively, these results suggest that animal models with AAV-Cre-GFP mediated replacement of γ2 subunit exhibit loss of diurnal patterns in γ2 expression and the activity of GABAergic input, and can be used to investigate the physiologic implication of BMAL1-controlled diurnal γ2 expression.

**Loss of diurnal γ2 rhythms on diurnal metabolism**. To examine the role of diurnal γ2 expression patterns in PVH neuron response to nutritional challenges, we bilaterally delivered either AAV-GFP vectors as controls (Fig. 6a, left two columns of panels) or AAV-Cre-GFP (Fig. 6a, right two columns of panels) to the PVH of γ2$^{flox/flox}$::γ2-TB$^{flox/flox}$ mice. Compared to control PVH neurons that showed robust c-Fos expression in response to fasting–refeeding, PVH neurons with Cre-mediated γ2 replacement exhibited minimal c-Fos induction (Fig. 6a, b). These results suggest that loss of diurnal γ2 rhythms blunted PVH neuron response to nutritional challenges during fast–refeeding.

AAV-Cre-GFP-injected γ2$^{flox/flox}$::γ2-TB$^{flox/flox}$ mice exhibited increased body weight within a few weeks after viral delivery

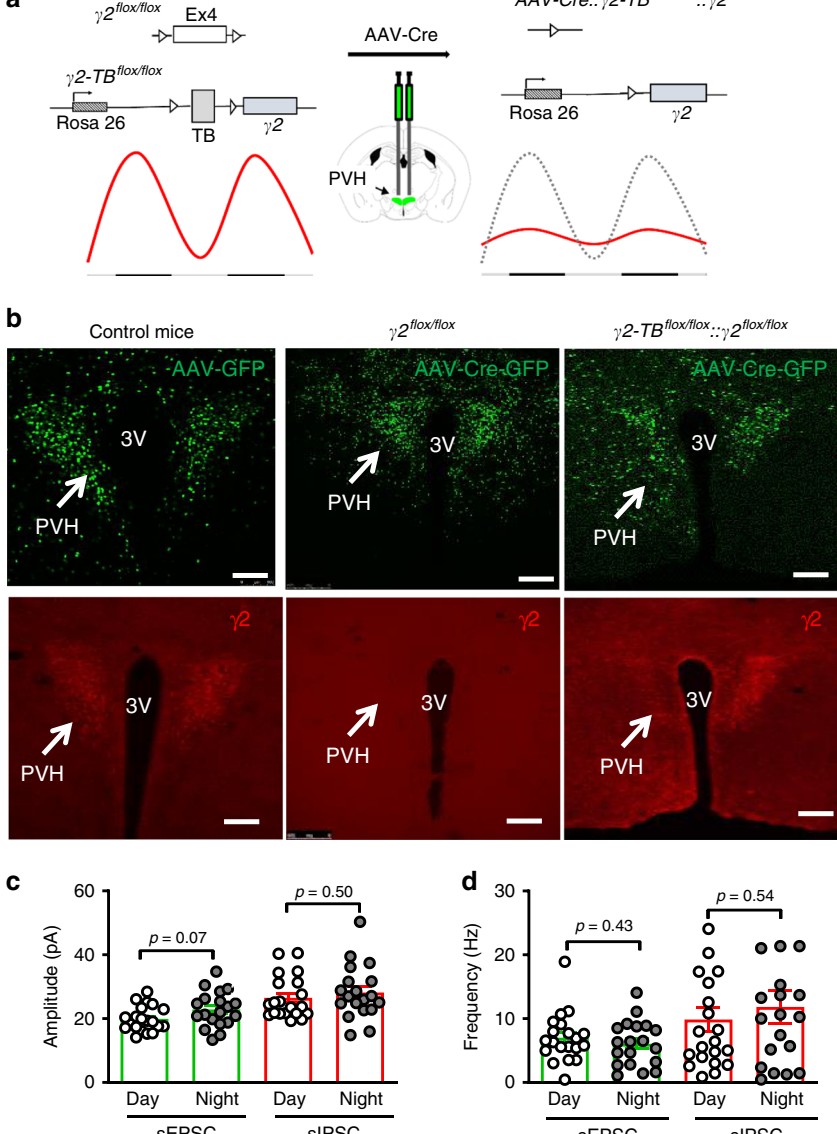

**Fig. 5 Generation of mice with loss of diurnal rhythms of γ2 subunit expression in adult PVH. a** Diagram showing delivery of AAV-Cre-GFP to the PVH of $\gamma2^{flox/flox}::\gamma2\text{-}TB^{flox/flox}$ to clamp γ2 expression at a constant level in adult mice. Cre-mediated deletion of the endogenous γ2 and expression of wild-type γ2 cDNA driven by the ROSA26 promoter lead to replacement of rhythmic expression of γ2 with a static expression level of γ2 driven by the ROSA26 promoter. **b** Validation of viral delivery to the PVH with GFP for vector delivery and red for immunostaining of γ2 expression. GFP expression (AAV-GFP or AAV-Cre-GFP) was verified in the PVH (top panels) and γ2 expression was verified by immunostaining (bottom panels): no expression of γ2 in Cre-mediated deletion (bottom middle, arrow) and re-expression by Cre-mediated γ2 expression driven by the ROSA26 promoter (bottom right, arrow), noticing consistency in the expression between Cre-GFP (top right) and γ2 (bottom right). **c, d** The amplitude (**c**, $n = 15$ each) and frequency ($n = 12$ for EPSCs and 14 for IPSCs) of sEPSCs and sIPSCs recorded during day and night periods from PVH neurons 3–4 weeks after AAV-Cre-GFP delivery. All data are represented as mean ± SEM and analyzed by unpaired Student's *t*-tests. TB transcription blocker. Scale bars: 200 μm, 3V third ventricle. Source data are provided as a Source Data file.

(Fig. 6c), with a net increase of 5 g within 7 weeks, compared to controls (Fig. 6d). In contrast to typical diurnal rhythms observed in energy expenditure (Fig. 6e) and food intake (Fig. 6g) in controls, mice with AAV-Cre-GFP injections had reduced diurnal rhythmicity in both measured (Fig. 6e, g). In AAV-Cre-GFP-injected mice, levels of energy expenditure (Fig. 6f) and amounts of feeding (Fig. 6h) were comparable between day and night periods, which is in contrast to typical diurnal rhythms observed in controls (Fig. 6f, h). When compared at 6–7 weeks after viral injections, AAV-Cre-GFP-injected mice showed slightly increased feeding (Fig. 6j), but significantly reduced energy expenditure (Fig. 6i), suggesting that the obesity

development is caused by reduced energy expenditure and increased feeding efficiency. Together, these data demonstrate that diurnal activity patterns of GABAergic input to PVH are required for normal diurnal patterns in metabolism and body weight regulation.

**PVH neuron responsiveness to stressful stimuli.** Blunted PVH neuron response to fast–feeding represents a common feature of HFD feeding and all animal models presented here. Since PVH neuron activity is orchestrated by a combination of synaptic and hormonal actions[35], we were also interested in whether the

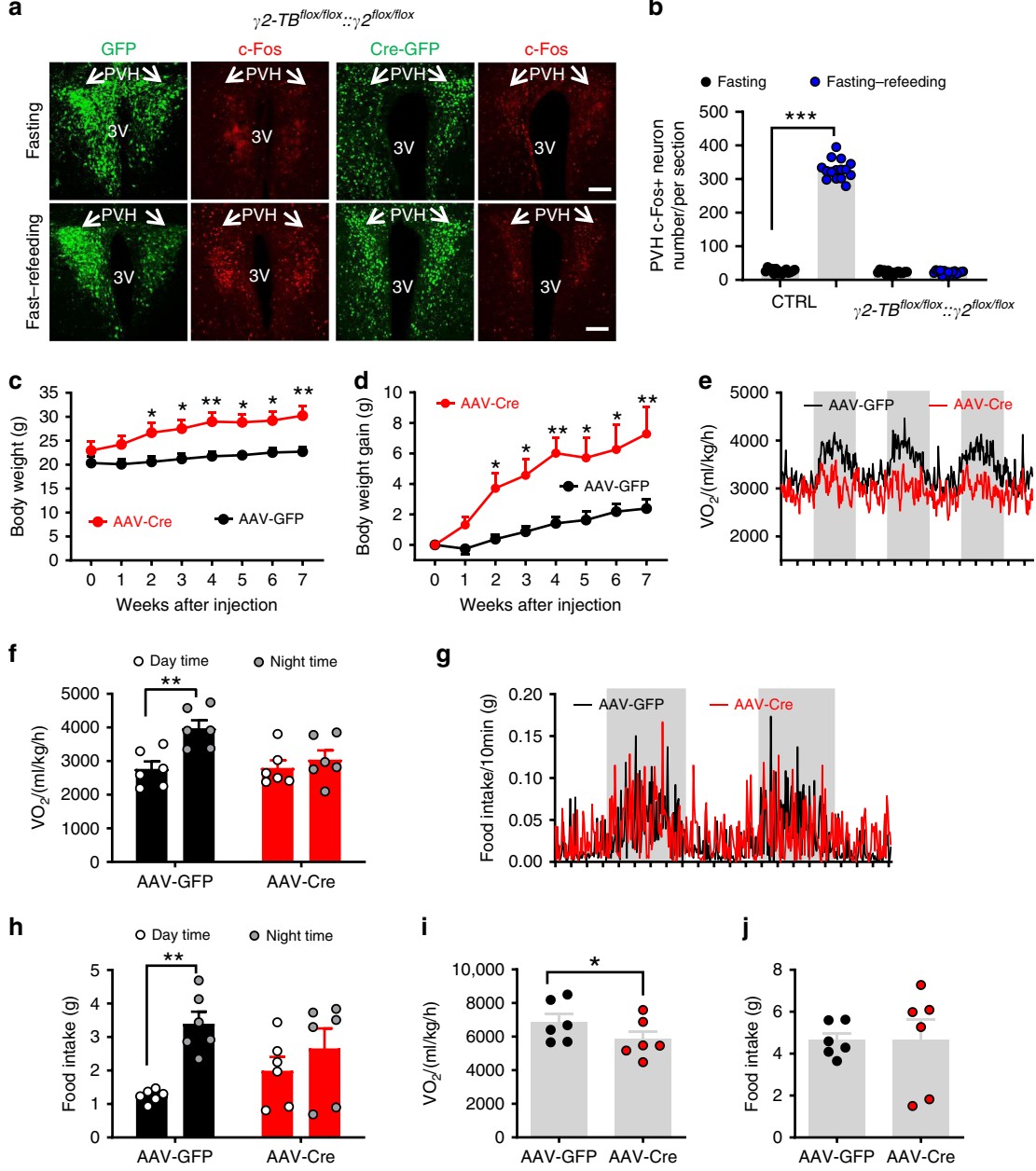

**Fig. 6 Diurnal rhythms of GABA-A γ2 subunit in the regulation of diurnal metabolism and body weight. a** c-Fos expression pattern in response to overnight fasting or overnight fasting with 2 h refeeding in $\gamma2^{flox/flox}::\gamma2\text{-}TB^{flox/flox}$ mice with PVH bilaterally injected with AAV-GFP (left panels) or AAV-Cre-GFP vectors (left panels). **b** Comparison of number of c-Fos-positive neurons in the PVH between fast and fast–refeeding in GFP- and Cre-injected mice (two-way ANOVA, $n = 15$/each, $F(3, 56) = 1438$, ***$p < 0.0001$, Control fasting vs fasting–refeeding; $p = 0.9984$, γ2 fasting vs fasting–refeeding). Arrows point to the PVH proper. Littermate control or $\gamma2^{flox/flox}::\gamma2\text{-}TB^{flox/flox}$ mice (8–10 weeks old, males) received injections of AAV-Cre-GFP to bilateral PVH and were used for body weight and feeding measurement using CLAMS. **c, d** Weekly body weight (**c**, two-way ANOVA, $n = 6$ each, $F(7, 80) = 2.682$, **$p = 0.0063$ for body weight at 7 weeks after injection between Control vs γ2 mice) and weekly body weight gain (**d**, two-way ANOVA, $n = 6$ each, $F(7, 80) = 8.042$, **$p = 0.0011$, body weight at 7 weeks after injection between Control vs γ2 mice) were followed. **e, f** Real-time energy expenditure $O_2$ consumption traces (**e**) and comparison in energy expenditure between day and night periods (**f**, two-way ANOVA, $n = 6$ each, $F(1, 20) = 3.541$, **$p = 0.0097$ GFP day vs night; $p = 0.883$ Cre day vs night). **g, h** Real time feeding traces (**g**) and comparison in feeding between day and night periods (**h**, two-way ANOVA, $n = 6$ each, $F(31, 20) = 3.380$, **$p = 0.0062$ GFP day vs night; $p = 0.656$, Cre day vs night). **i, j** Comparison of average daily $O_2$ consumption (**i**, unpaired two-tailed Student's $t$-test, $n = 6$ each, $t = 2.231$, d.f. $= 10$, *$p = 0.0498$) and food intake (**j**, $n = 6$ each). All data are represented as mean ± SEM. Scale bar: 100 µM. Shaded bars represent dark periods. Source data are provided as a Source Data file.

observed loss in response to fast–refeeding may reflect a general role for BMAL1-driven program in PVH neuron function. To examine this possibility, we monitored in vivo PVH neuron activity using fiber photometry in response to water spray, which produces a controlled stress stimulus to animals. We delivered AAV-Flex-GCaMP6m to the PVH of *Sim1-Cre* mice (Fig. 7a). To

control for potential interference from the background fluorescence presence in GCaMP recording, we also used *Sim1-Cre::Ai9* mice with tdTomato expression in the PVH (Fig. 7b). Indeed, we documented that a single water spray elicited rapid and strong activation of PVH Sim1 neurons (Fig. 7c, i). However, responsivity to the same stimulus was greatly diminished in PVH Sim1

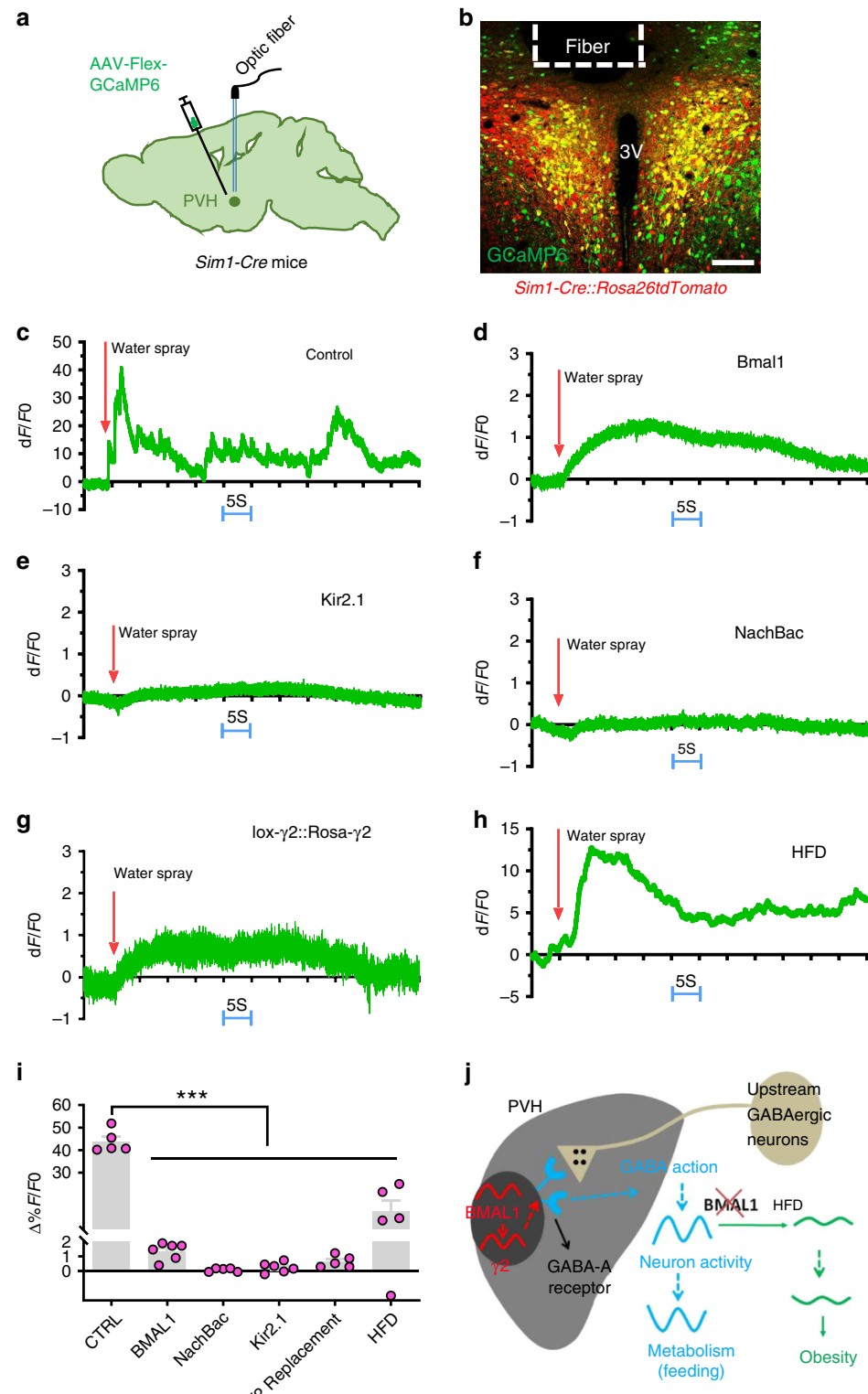

neurons with BMAL1 deletion (Fig. 7d, i), or targeted expression of either Kir2.1 (Fig. 7e, i) or NachBac (Fig. 7f, i), or with γ2 replacement (Fig. 7h, i). Interestingly, the response was also diminished in the same group of mice after 4-week HFD feeding (Fig. 7h, i), demonstrating the effect of HFD in disrupting responsiveness of PVH neurons. These results suggest that the BMAL1-mediated program in the PVH may control a general function of PVH neuron responsiveness.

## Discussion

Given the importance of rhythmic metabolism and feeding to body weight regulation[6,36], it is critical to identify brain sites that govern diurnal patterns in metabolism, feeding, and locomotion. Here we identify the PVH as a key brain site that is required for diurnal patterns in metabolism, feeding, and locomotion through rhythmic expression of BMAL1 and its regulated GABAergic inputs. Importantly, the expression of BMAL1 is required for

**Fig. 7 Responsiveness of PVH Sim1 neurons to acute stress. a, b** A diagram showing delivery of AAV-Flex-GCaMP6m viral vectors to bilateral PVH of *Sim1-Cre* mice (**a**) and representative expression of GCaMP6m (green) and tdTomato in the PVH after delivery of AAV-Flex-GCaMP6m to *Sim1-Cre::Ai9* mice (**b**). **c, e, f, h** *Sim1-Cre* (or *Sim1-Cre::Ai9*) male mice received stereotaxic delivery of AAV-FLEX-GCaMP6m (**c, h**), a mixture of AAV-FLEX-GCaMP6m and AAV-DIO-Kir2.1-P2A-dTomato vectors (**e**), a mixture of AAV-FLEX-GCaMP6m and AAV-EF1a-FLEX-EGFP-P2A-mNachBac vectors (**f**). **d, g** *Bmal1^flox/flox^* mice (**d**) or *γ2^flox/flox^::γ2-TB^lox/lox^* mice (**g**) receiving a mixture of AAV-FLEX-GCaMP6m and AAV-Cre-GFP (to bilateral PVH at 7–8 weeks of age with optic fiber implantation targeting PVH neurons). There mice were then used to monitor acute activity responses to a single water spray with sprayer positioned at a fixed distance toward mouse head (arrows indicating onset of water spray). **c–g** A representative trace showing responses of PVH Sim1 neurons to water spray in the animal model indicated. **h** A representative response of the same mice used in **c** after 6 weeks on HFD. **i** Comparison in activation of PVH neurons to water-spray among mouse groups. Data are represented as mean ± SEM. One-way ANOVA, $n = 5$ or 6, ***$p < 0.0001$, Control vs Bmal1, vs NachBac, vs Kir2.1, vs γ2, or vs HFD. All mice were fed chow unless otherwise noted. **j** A simplified diagram showing the proposed mechanism based on the presented results. Diurnal BMAL1 expression pattern elicits diurnal patterns in GABAergic inputs through regulation of γ2 subunit, which maintains a diurnal rhythm of PVH neurons activity and metabolism. HFD diminishes diurnal rhythms of PVH neuron activity, likely through blunting neuron responsiveness and promotes obesity through disruption of diurnal rhythms in metabolism. All data are presented as mean ± SEM. Source data are provided as a Source Data file.

PVH neurons to mount an adaptive response in neuron activation to the metabolic change, i.e. fast–refeeding, a condition frequently occurring during normal physiology. The adaptive response to metabolic changes plays a critical role in diurnal pattern generation of metabolism because our models with PVH neuron activity clamped at high (i.e. NachBac), or low (i.e. Kir2.1) levels, both with loss of response to metabolic changes, develop obesity with reduced diurnal rhythms in metabolism. It is worth noting that HFD feeding caused a similar effect in disrupting PVH neuron responsiveness, directly implicating loss of PVH neuron responsiveness in mediating DIO. PVH neuron activity is known to exhibit a diurnal pattern[28], which is conceivably orchestrated by dynamic changes in synaptic inputs and hormonal actions as well as cell-autonomous changes. Thus, loss of PVH neuron adaptive responsiveness in our models will most likely disrupt diurnal activity patterns of PVH neurons, ultimately leading to dampened diurnal patterns of metabolism and feeding and obesity. Taken together, as diagrammed in Fig. 7i, these data support that BMAL1-driven diurnal PVH neuron activity rhythm through modulating GABAergic input represents a major part of brain mechanism underlying diurnal patterns in metabolism and DIO.

Despite the importance of the PVH in body weight regulation, the understanding on PVH function is largely limited to static control of feeding and energy expenditure[15]. Also how acute feeding effects induced by transient changes in PVH neuron activity can be translated to body weight regulation is not clear[37–39]. Our results suggest that it is not the absolute neuron activity level, but the PVH neuron responsiveness, governs feeding efficiency and body weight. This is best exemplified by our models with PVH neuron activity clamped at a high or low level, which both disrupted neuron responsiveness, reduced diurnal rhythms in metabolism, increased feeding efficiency, and caused obesity. While the obesity phenotype resulted from PVH neuron inhibition is consistent with the known role of these neurons in feeding, the obesity phenotype in mice with an increased level of PVH neuron activity (i.e. NachBac) is opposite to what would be predicted from the known effect of transient activation of PVH neurons in acute feeding inhibition as shown previously[38]. One potential explanation may lie in the difference between acute vs chronic alterations in PVH neuron activity. Despite previous studies implicating acute activation of various subsets of PVH neurons in feeding inhibition[39,40], an animal model on these neurons presented with a lean body weight phenotype, the presumptive phenotype from reduced feeding, has yet to be reported. Although further studies are warranted, this phenomenon may indicate that, contrary to the prediction from PVH neuron inhibition in obesity development, activation of these neurons is not sufficient to reduce body weight. Supporting this, while

inhibition of proopiomelanocortin (POMC) neurons, one key upstream neurons of PVH melanocortin receptor 4 (MC4R) neurons, causes massive obesity, activation of these neurons fails to produce a lean phenotype[41]. Nevertheless, our results highlight an importance of PVH neuron responsiveness, dynamic changes in PVH neuron activity and diurnal patterns in metabolism in body weight regulation. Of note, our data on the causal role for the PVH in diurnal rhythms of metabolism are supported by non-rhythmic feeding and energy expenditure in mice with adult lesions of PVH neurons[42], and dependence on timing in GABAergic action on PVH neurons in physiology[43,44]. Collectively, our study reveals the PVH as a key site to mediate the clock gene function on diurnal rhythms of feeding and metabolism[6–8,45]. Given the diverse roles of PVH neurons, the diurnal rhythmicity of PVH neurons may also play an important role in maintaining other physiological functions.

PVH neurons are known to regulate feeding as demonstrated by previous studies on prominent effects on feeding with optogenetic or chemogenetic manipulation of the activity of these neurons[40,46] or in obese animal models with altered PVH neuron function[42,47]. It is intriguing that obesity in all animal models reported here was not associated with a significant change in feeding but rather with reduced energy expenditure. The major reason may be due to the fact that our CLAMS measurement was performed at an early time point, weeks 2 and 3 after viral delivery when there was no or litter body weight difference. Notably, earlier studies in animal models on PVH neurons reporting hyperphagia measured food intake at a time point with considerable obesity development[39,45] and it is conceivable that the feeding difference is amplified. In contrast, when food intake is measured before obesity development in animal models with PVH-specific manipulations of MC4Rs or glutamate release, food intake is not obviously different[48,49]. In particular, an obesity-dependent effect of food intake difference has been nicely demonstrated in PVH neuron-specific manipulation of MC4Rs[49]. Given the tight association in diurnal patterns between feeding and energy expenditure, the PVH may regulate food intake and energy expenditure simultaneously. The animal models presented here all exhibited a more pronounced disruption in diurnal patterns of energy expenditure than feeding, specifically, more reduction in energy expenditure than feeding during dark periods, suggesting a higher feeding efficiency, which may drive obesity development.

The neural basis for diurnal rhythms in PVH neuron activity may be complex and may include dynamic changes in cell-autonomous rhythms, and presynaptic input and hormonal action in response to internal and external perturbations[50,51]. The cell-autonomous rhythmic changes may be orchestrated by clock genes, as demonstrated by our results from BMAL1 deletion in

the PVH and its regulation of downstream genes, including GABA-A receptors. The effects on PVH neuron activity by external perturbations are reflected by a high level of sensitivity of PVH neurons to stress stimuli (e.g. water spray), which is likely mediated through rapid neurotransmission in synaptic inputs[52,53]. It is conceivable that PVH neurons are also sensitive to daily internal and external perturbations such as changes in light conditions, existing or perceived dangers, states in hunger, or even blood pressure[35,54,55]. Thus, the responsiveness of PVH neurons likely contribute significantly to shaping the diurnal activity pattern of these neurons. Of note, our results show that PVH neuron responsiveness was greatly attenuated by HFD feeding, which is known to be associated with disrupted diurnal activity pattern of these neurons. The underlying reasons for the blunted responsiveness is unknown but likely involve changes in synaptic function as synaptic inputs to the PVH are known to exhibit a high level of reorganization activity[33,55]. In particular, HFD is known to be able to induce changes in synaptic reorganization by hormonal action or PVH-projecting pathways[56–58]. Nonetheless, these observations demonstrate an interactive role for both cell-autonomous rhythmic changes and extracellular inputs in orchestrating the diurnal activity pattern of PVH neurons. Given the known effect of HFD on reducing diurnal rhythms in metabolism[9], our current findings suggest that HFD-induced blunting PVH neuron responsiveness contributes to DIO. Thus, the PVH functions as a converging site for both circadian genes and HFD feeding on metabolism.

Our results identified GABAergic input as one of major pathways in BMAL1-mediated cell-autonomous diurnal changes. These changes may contribute significantly to a low level of PVH neuron activity during night periods. One of major sources of GABAergic inputs may be from the Arc and lateral hypothalamus as our previous results demonstrate that disruption of GABA release from these sites substantially reduces GABAergic input to PVH neurons and blunts diurnal rhythms in feeding and energy expenditure[20]. Since AgRP neurons are known to send direct GABAergic inputs to the PVH and the activity of these neurons increases during night periods[16,19], the increase in PVH γ2 expression may cause a synergistic effect on amplifying GABAergic action onto PVH neurons, as shown by increases in both sIPSC amplitude and frequency during the night period. In addition to AgRP neurons, other non-AgRP GABAergic neurons in the hypothalamus, including arcuate tyrosine hydrolase (TH) and Rip-Cre neurons[21,59], known to project directly to the PVH, may also contribute to the increased evoked IPSC frequency in PVH neurons. Consistently, ablation of Arc neurons that express leptin receptors, most of which are GABAergic neurons[22], leads to disruption of diurnal patterns of feeding and locomotion[23], suggesting a potential involvement of leptin action in shaping PVH neuron responsiveness. Importantly, loss of diurnal γ2 expression by replacing endogenous γ2 with ROSA26-mediated expression led to loss of diurnal patterns of IPSCs between day and night periods, providing direct evidence for dynamic γ2 expression in modulating neuron activity. In line with this, previous studies suggest a role for GABAergic input in controlling PVH neuron activity[43,52]. The diurnal expression of γ2 may be part of diurnal rhythmic function of GABA-A receptors, as diurnal expression of other GABA-A receptor subunits has been previously reported[27]. The increased GABAergic action during night periods may represent a general phenomenon of brain activity given that levels of whole-brain GABA content elevates during the night[26]. Thus, our data show that diurnal rhythmicity in metabolism is mediated by PVH neurons through concurrent enhancements in both pre- and postsynaptic GABAergic actions.

Unexpectedly, we observed that disruption of rhythmic γ2 expression in the PVH also dampened the increase in nocturnal sIPSC frequency, an indicator for presynaptic GABA release. One possibility might be the disrupted feeding rhythms. PVH circadian genes including Bmal1 are efficiently entrained by feeding[24]. Although it remains to be determined, the entrainment may be mediated by GABA release from the "first-order" neurons that project to the PVH, and that feeding may modulate GABA release from these neurons, as arcuate GABAergic neurons are known to respond to an array of hormonal factors tuned by feeding status[60]. Candidate neurons include those expressing leptin receptors, since arcuate GABAergic leptin receptor neurons are implicated in mediating diurnal feeding patterns[22,23]. Similarly, other neurons known to sense feeding-related signals such as ghrelin, insulin, leptin, and glucose may also be significant contributors[19,21,60]. Alternatively, the γ2 subunit enhances the efficacy of GABA, thereby producing longer duration of open GABA-A receptors with greater currents[61].

Previous results show that deletion of γ2 in the PVH mediated by Sim1-Cre transgene causes a mild reduction in nocturnal energy expenditure and feeding[20]. Here we showed that Sim1-Cre-mediated replacement of the endogenous γ2 expression with one driven by the Rosa26 promoter, although capable of rescuing body weight effect caused by Sim1-Cre-mediated deletion of γ2 (ref. [20]), showed normal diurnal patterns in feeding and energy expenditure. However, the same manipulations mediated by AAV-Cre-GFP delivery in adult mice resulted in notable disruption in diurnal metabolism and obesity. These contrasting results may suggest powerful functional compensations for Sim1-Cre-mediated genetic manipulations during embryonic and/or early postnatal development. A similar developmental compensation has also been observed for lesion of AgRP neurons[62] and loss of glutamate release from PVH neurons[48]. Notably, a striking difference is that Sim1-Cre-mediated γ2 deletion causes no effects on PVH neuron survival whereas AAV-Cre-GFP-mediated adult brain deletion causes excitotoxicity and neuronal death, suggesting that the compensation for loss of γ2 subunit may be mediated through balancing excitatory inputs. Further investigation into the nature of this observed developmental compensation may identify novel mechanisms that underlie neural plasticity.

In the current study, we focused on the function of all PVH neurons since the entire PVH exhibits abundant diurnal patterns of expression in both GABA-A receptor γ2 subunit (this study) and BMAL1 (ref. [24]). However, it is unknown whether distinct PVH neuron subsets play a differential role in rhythmic regulation of feeding vs energy expenditure. Previous results show that some groups of PVH neurons selectively regulate feeding or energy expenditure[40,47,48], arguing for a possibility of differential regulation. Future studies on relative contributions of individual subsets of PVH neurons will identify key PVH neurons for diurnal patterns in energy expenditure, feeding, and/or locomotion. In summary, our study identifies BMAL1-driven PVH neuron responsiveness as one underlying mechanism of diurnal patterns of metabolism, feeding, and locomotion. Defective rhythms in metabolism by loss of PVH neuron responsiveness or rhythmic GABAergic input to the PVH cause obesity, demonstrating a novel mechanism underlying PVH neurons in obesity development. Importantly, HFD feeding reduces PVH neuron responsiveness, implicating PVH neurons in contributing to HFD-induced blunting effects on diurnal rhythms in metabolism. Our findings collectively identify PVH as a non-SCN brain l site that orchestrates diurnal rhythms in metabolism, feeding, and locomotion, which may represent an alternative and effective strategy against obesity.

## Methods

**Animal care.** Mice were housed at 21–22 °C with a 12 h light/12 h dark cycle with standard pellet chow and water provided ad libitum. Animal care and procedures

were approved by the University of Texas Health Science Center at Houston Institutional Animal Care and Use Committee. $Bmal1^{flox/flox}$ mice were obtained from the Jax lab[63] and verified with Cre-mediating deletion of $Bmal1$ expression. $\gamma2^{flox/flox}$ and $Sim1$-Cre mice were used and verified previously[20,25]. To generate $\gamma2^{flox/flox}::\gamma2\text{-}TB^{flox/flox}$ mice, wild-type $\gamma2$ coding sequence was replaced right after a ROSA26 promotor using homologous recombination. To achieve that the expression is Cre-dependent, a transcript blocker (TB) flanked by two loxP site was placed right before the $\gamma2$ sequence. Cre-mediated removal of TB induces the expression of wild-type $\gamma2$ subunit. A Chicken B-actin/CMV promoter was inserted right after the ROSA26 promotor to ensure sufficient expression levels. The following breeding pairs were maintained to generate study subjects: 1. $\gamma2^{flox/flox} \times \gamma2^{flox/flox}$; 2. $Sim1^{cre/+}::\gamma2^{flox/flox}::\gamma2\text{-}TB^{flox/flox} \times \gamma2^{flox/flox}::\gamma2\text{-}TB^{flox/flox}$. In addition, $Sim1$-Cre mice were bred to Ai9 reporter mice to generate $Sim1$-Cre:Ai9 mice for electrophysiological recording and visualization of PVH neurons. Genotyping PCR primers used were for Sim1-Cre mice: Sim1-Up 5'-CCGAGTGTGATC TCTAATTGAAAG-3', Sim1-Down 5' GTCGTGAGTCGTTGATGATGGCT-3', and Sim1-Reverse 5' GGTGTACGGTCAGTAAATTGGACACCT-3'; for $\gamma2^{flox/flox}$ mice: $\gamma2$-up Forward 5'-GTCATGCTAAATATCCTACAGTGG-3', 2-up Reverse 5'-GGATAGTGCATCAGCAGACAATAG-3', and $\gamma2$-down Forward 5'-GAATGT TGATATGTAACCAAG-3'; for $\gamma2\text{-}TB^{flox/flox}$ mice: Rosa-tdTomato WT 5'-CCGA AAATCTGTGGGAAGTC-3', Rosa-tdTomato Com 5'-AAGGGAGCTGCAGTGG AGTA-3'; Rosa-$\gamma2$ Forward 5'-CAAGATCATGGAGGCTGTATC-3' and Rosa-$\gamma2$ Reverse 5'-GTAAGTCTGGATGGTGAAGTAG-3'; and for $Bmal1^{flox/flox}$ mice: Bmal-forward 5'-ACTGGAAGTAACTTTATCAAACTG-3' and Bmal-Reverse 5'-CTGACCAACTTGCTAACAATTA-3'.

**Surgeries and viral constructs**. Adult male mice were placed on a stereotaxic frame (David Kopf Instruments) under the anesthetized condition with ketamine. AAV-GFP and AAV-Cre-GFP vectors were purchased from the viral core facility of the University of Pennsylvania. The AAV-EF1a-FLEX-EGFP-P2A-mNachBac vector was constructed and provided in the lab of Dr. Benjamin Arenkiel[32]. The AAV-EF1a-DIO-Kir2.1-P2A-dTomato vector was obtained from Dr. Mingshan Xue[30]. Viral vectors were delivered through a 0.5 μL syringe (Neuros Model 7000.5 KH, point style 3; Hamilton, Reno, NV, USA) mounted on a motorized stereotaxic injector (Quintessential Stereotaxic Injector; Stoelting, Wood Dale, IL, USA) at a rate of 10 nl/min. Viral preparations were titered at ~$10^{12}$ particles/ml. The vectors were stereotaxically injected into bilateral PVH (50–100 nl per site) with following coordinates: anteroposterior (AP): −0.5 to −0.6 mm; mediolateral (ML): ±0.2 mm; dorsoventral (DV): −4.8 mm). Mice were allowed to recover for a week, after which they were used for experiments. AAV-GFP injections were used as a control group.

**In vivo fiber photometry experiments**. GCaMP6m expression in the PVH achieved by specific delivery of the AAV-FLEX-GCaMP6m virus to the PVH of $Sim1$-Cre mice, $Sim1$-Cre::Ai9 reporter mice, or by specific delivery of a mixture of the virus and AAV-Cre-GFP to the PVH and optic fiber implantation targeting PVH neurons were used for the in vivo fiber photometry $Ca^{2+}$ imaging studies[64]. The GCaMP6m virus was provided by the Baylor NeuroConnectivity Core. The experiments were conducted at least 4 weeks after the surgery. All fiber photometry was conducted using a Doric Lenses setup, with an LED Driver controlling two connectorized LEDs (405 and 465 nm) routed through a five port Fluorescence MiniCube (order code: FMC5_AE(405)_AF(420-450)_E1(460-490)_F1(500-550) _S) to deliver excitation light for calcium-independent and -dependent signals to the implanted optic fiber simultaneously. Emitted light was received through the MiniCube and split into two bands—420–450 nm (autofluorescence—calcium-independent signal) and 500–550 nm (GCaMP6 signal—calcium-dependent signal). Each band was collected by a Newport 2151 Visible Femtowatt Photoreceiver module (photometer) with an add-on fiberoptic adaptor. Output analog signal was converted to digital signal through the fiber photometry console and recorded using the "Analog-In" function on Doric Studios (V4.1.5.2). Mice were acclimated to the behavioral chamber for at least 15 min prior to the beginning of each testing session. After baseline recording for 10 s, water-spray was started by spraying one time with water toward the head of mice with a sprayer and the recording will continue for about 2 min.

Data were acquired through Doric Studios V4.1.5.2. and saved as comma-separated files (header was deleted) at a sampling rate of 1 kS/s. An in-house script was written in R (V3.4.4. "Someone to Lean On", packages ggplot2, reshape, zoo, plyr, viridis, scales) was used to calculate the baseline fluorescence ($F0$) using linear regression across a chosen period of recording. The change in fluorescence ($dF$) was then determined from the residuals and multiplied by 100 to arrive at % $dF/F0$. A sliding median with a window 51 data points in width was used to smooth data. For water spray, the baseline was calculated from 10 s prior to stimulus onset to 20 s after stimulus onset.

**Brain slice electrophysiological recordings**. Coronal brain slices (280–300 μm) containing the paraventricular nucleus of the hypothalamus (PVH) were cut in ice-cold artificial cerebrospinal fluid (aCSF) containing the following (in mM): 125 NaCl, 2.5 KCl, 1 MgCl₂, 2 CaCl₂, 1.25 NaH₂PO₄, 25 NaHCO₃, and 11 D-glucose bubbling with 95% O₂/5% CO₂, either from WT mice or mice that had received

stereotaxic injections of AAV-GFP (AAV-Control), AAV-Cre-GFP, AAV-EF1a-FLEX-EGFP-P2A-mNachBac or AAV-EF1a-DIO-Kir2.1-P2A-dTomato to bilateral PVH at least 3 weeks prior to the recording. In brief, slices containing the PVH were immediately transferred to a holding chamber and submerged in oxygenated aCSF, and maintained for recovery for at least 1 h at 32–34 °C before transferring to a recording chamber. Individual slices were transferred to a recording chamber mounted on an upright microscope (Olympus BX51WI) and continuously perfused (2 ml/min) with ACSF warmed to 32–34 °C by passing it through a feedback-controlled inline heater (TC-324B; Warner Instruments). Cells were visualized through a 40× water-immersion objective with differential interference contrast optics and infrared illumination. Whole-cell recordings were made from neurons within the sub-region of dorsal medial part of the PVH. Patch pipettes (3–5 MΩ) were filled with a K⁺-based low Cl⁻ internal solution containing (in mM) 125 K-gluconate, 10 EGTA, 5 HEPES, 2 KCl, 5 MgATP, 0.3 NaGTP (pH adjusted to 7.3 with KOH, ~290 mOsm) for voltage-clamp recordings of the spontaneous post-synaptic currents (sIPSC and sEPSC) with recording voltage at −40 mV[33]. For current-clamp recordings, pipettes were filled with a K⁺-based internal solution containing (in mM) 126 K-gluconate, 10 NaCl, 10 EGTA, 1 MgCl₂, 2 ATP, 0.1 GTP (pH 7.3 adjusted with KOH; ~290 mOsm).

To study the eIPSCs in labeled PVN neurons, synaptic currents were evoked by electrical stimulation (0.1 ms; 0.3–0.8 mA; 0.1 Hz) through a bipolar tungsten electrode connected to a stimulator (FHC, Inc. Bowdoin, ME). The resistance of the pipette was 4–8 MΩ when filled with the internal solution containing the following (mM): 130 K- gluconate, 10 HEPES, 10 EGTA, 1 MgCl₂, 1 CaCl₂, and 4 ATP-Mg (pH 7.3, with 1 M KOH, ~290 mOsm). The tip of the stimulating electrode was placed 200–300 μm away from the recorded neuron. $N$-ethyl bromide quaternary salt (QX-314) (10 mM) and GDP-β-s (1 mM) were included in the pipette solution to block the Na⁺ current and possible postsynaptic effect in these voltage-clamp experiments. On the basis of the optimal reversal potentials determined for bicuculline-sensitive IPSCs, the eIPSCs were recorded at a holding potential of 0 mV. pCLAMP10 was used for data collection.

**Food intake, O₂ consumption, locomotion activity**. Food intake, O₂ consumption, and locomotion activity levels were measured using indirect calorimetry. Mice were individually housed in chambers of a CLAMS (Columbus Instruments). Mice were given ad libitum access to normal chow food and water. Mice were acclimatized in the chamber for at least 2 days prior to data collection. The measurement was performed at various time points after delivery as indicated in each study. Measurements on O₂ consumption, feeding, and locomotion were collected continuously during the whole measurement period. Data were prepared by averaging for 24 h, or for day and night periods. For voluntary home cage movements, running wheels were placed in home cage with individually housed mice and the wheel running activity was measured for at least 2 weeks.

**Immunostaining and c-Fos counting**. For immunohistochemistry studies, we used the previously descried protocol[20]. Brain sections were prepared from control or $Sim1$-Cre::$\gamma2^{flox/flox}::\gamma2$-$TB^{flox/flox}$ male adult mice (8–10 weeks) after cardiac perfusion at ZT0, 6, 12, and 18. For validation of AAV-GFP and AAV-Cre-GFP mediated $\gamma2$ subunit deletion and replacement, brains were harvested at the end of studies with cardiac perfusion, except mentioned otherwise in the text. Primary antibody against $\gamma2$ (1:100, NB300-190, Novus Biologicals), BMAL1 (1:200, NB100-2288, Novus Biologicals), c-Fos (1:1000, #2250, Cell Signaling Technology), and cleaved caspase-3 (1:300, #9664, Cell Signaling) were used. The sections were visualized with Alexa Fluor 488® or Alex Fluor 594® (1:200, Jackson immunoR-esearch Laboratories, Inc.). The signal was captured and imaged with a TCS SP5 confocal microscope (Leica, Nussloch, Germany).

For c-Fos counting, five sections that contain the PVH at matched anteroposterior levels were chosen from each mouse. Within the PVH boundary, all c-Fos-positive cells with bright and clear oval nuclei profiles were counted under the fluorescent microscope with the 200× objective. The number from all sections were averaged and analyzed in all animals.

**Luciferase assay**. Neuro 2A cells were cultured in Dulbecco's modified Eagle's medium supplemented with 10% fetal bovine serum (Atlanta), 100 IU/ml penicillin and 100 ng/ml streptomycin. To measure BMALl and CLOCK activity on the gamma 2 promoter, Neuro 2A cells were seeded into a 24-well plate overnight and then transfected with 500 ng of the Gamma2-luciferase reporter plasmid combined with 10 ng of pRL-SV40 (Promega), 50 ng of pcDNA3.1-Bmall and 50 ng pCMV10/3xflag-Clock plasmids or the control empty plasmids, according to the Lipofectamine 2000 protocol (Invitrogen). Forty hours post-transfection, the cells were lysed and the luciferase activity was measured using the Dual-Luciferase® Reporter Assay System (Promega) according to the manufacturer's instruction.

**Chromatin immunoprecipitation**. Chromatin immunoprecipitation was carried out using the Upstate ChIP assay kit. Briefly, hypothalamus was collected from wild-type mice at ZT7. Tissues were washed with cold phosphate-buffered saline (PBS) and homogenized, followed by formaldehyde (1% final concentration) incubation for 10 min at 37 °C. After 2× washes with ice-cold PBS, the pellets were dissolved with 200 μl SDS lysis buffer with protease inhibitor cocktail (Roche

Applied Science), and sonicated to obtain small DNA fragment between 200 and 500 bp. The lysates was centrifuged for 10 min at 13,000 r.p.m. at 4 °C. The supernatant was diluted 10-fold in ChIP dilution buffer with protease inhibitors cocktail, and 1% diluted cell lysis was kept as a positive control. Cell supernatant was pre-cleared by Salmon sperm DNA/protein A agarose slurry, and incubated with the Bmal1 antibody (1:200, Abcam #ab3350) overnight at 4 °C. Samples that were incubated without the Bmal1 antibody served as the negative control and incubated with Salmon sperm DNA/protein A agarose slurry for 1 h. The chromatin complexes were sequentially washed in low salt, high salt, LiCl salt, and TE buffers. The protein/DNA complex was eluted in an SDS elution buffer (1% SDS, 0.1 M NaHCO$_3$). The crosslink between protein and DNA was reversed. The protein/DNA complex was treated with Proteinase K. DNAs were purified by phenol/chloroform extraction, ethanol precipitation, and re-suspended in 50 μl of sterile water. DNAs were used as templates for PCR amplification and 1% genomic DNA was used as the positive control (input). The primers used were listed as following: site 1-F-AAGAGTGCTGGCAAGAATTCAC and site 1-R- CTGGTTCT GTTGTGTGGGAAA (expected product of 143 bp), or site 2-F-CCCTTCAATTT ACATGCACCA and site 2-R- CTTTCCTGTAGCAGCAGACTAGG (expected product of 232 bp). The PCR products were visualized with agarose gel electrophoresis.

**Statistics**. Initial data were collected in Microsoft Excel 2020. GraphPad Prism 8.0 (GraphPad Software, Inc., La Jolla, CA, USA) was used for all statistical analyses and construction of graphs. Two-way ANOVA or one-way ANOVA followed by Tukey's or Sidak's multiple comparison post hoc tests were used for group comparisons. Single variable comparisons were made by unpaired two-tailed Student's $t$-tests.

## Data availability

All available data are presented in figures and Supplementary Figures. Source data are provided with this paper.

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

## Acknowledgements

This study was supported by NIH R01 DK114279 and R21 NS108091 to Q.T.; R01DK117281, P01DK113954, and R01DK101379 to Yong Xu; R01DK109934 to B.R.A. and Q.T.; R01DK114037 to K.E.-M.; R01MH117089 and McKnight Foundation to M.X. We also acknowledge the Neuroconnectivity Core funded by NIH IDDRC grant 1 U54 HD083092 and Baylor College of Medicine Gene Vector Core for providing AAV vectors. Q.T. is the holder of Cullen Chair in Molecular Medicine at McGovern Medical School.

## Author contributions

E.R.K. conducted the major experiments and analyzed data related to the γ2 and NachBac mouse models. Yuanzhong Xu conducted the major experiments and analyze data related to the Kir2.1, part of γ2 mouse models and fiber photometry experiments. R. M.C. analyzed fiber photometry data. Y.L. performed electrophysiological experiments and analyzed the data related to diurnal EPSCs and IPSCs as well as the NachBac animal model. J.T. performed electrophysiological experiments and analyzed the data related to the Kir2.1 model. D.-P.L. performed electrophysiological experiments and analyzed the data related to generation of the ROSA-LSL-γ2 animal model. Y.Y. and Yong Xu performed γ2 CHIP and luciferase assays and analyzed the data. R.V.D., A.R.-L., and K.E.-M. measured circadian movements. Z.-L.C., A.R.-L., M.X., K.E.-M., and B.R.A. provided essential reagents; Q.T. generated the ROSA-LSL-γ2 mouse strain, perceived and designed the experiments, and wrote the manuscript with essential inputs from Yong Xu, M.X., B.R.A., and K.E.M.

## Competing Interests

The authors declare no competing interests.
