## [Peer Review File · Nature Communications]

Peer Review Information

Manuscript title: Paraventricular Hypothalamus Mediates Diurnal Rhythms of Metabolism

Corresponding author name(s): Dr Qingchun Tong

Editorial notes:

Transferred manuscripts

This manuscript has been previously reviewed at another journal that is not operating a transparent peer review scheme. This document only contains reviewer comments and rebuttal letters for versions considered at *Nature Communications*.

Redactions – unpublished data

Reviewer comments & decisions:

Reviewer comments, first version:

Reviewers' comments:

Reviewer #1 (Remarks to the Author):

The manuscript by Kim et al. addresses the neural basis of metabolic diurnal rhythms, an understudied topic. They employed state-of-the-art genetic approaches to determine the role of GABAergic inputs into the PVH in diurnal rhythms of energy expenditure, locomotion and food intake. They found that deleting the clock gene BMAL1 in the PVH abolished the diurnal rhythm of energy expenditure. They then tried to demonstrate that BMAL1 controls metabolic rhythms via regulating the rhythms of GABAergic inputs into PVH neurons by examining GABA-A receptor $\gamma 2$ expression, altering PVH neuronal activities, and abolishing diurnal rhythms of $\gamma 2$ expression. Findings reported in the manuscript are potentially interesting; however, scientific rigor in some experiments needs to be strengthened and some results are in conflict with the literature.

1. Stereotaxic injection of AAV in Figure 1 and Figure 5 is not specific to the PVH, which makes it difficult to conclude that BMAL1 and $\gamma 2$ in the PVH regulate metabolic rhythms. For example, AAV-Cre-GFP infection is present in the subparaventricular zone and the periventricular hypothalamic nucleus in addition to the PVH (Fig. 1d). In fact, BMAL1 expression in these two structures is higher than in the PVH (Fig. 1b). Thus, the observed phenotypes could be a result of BMAL1 deletion in these two

structures.

2. Figures 1o-q: c-Fos immunoreactivity shown in the figures is not consistent with the quantification. The images show that there are many c-Fos positive cells in fasted mice and Bmal1 mutant mice, while the bar graph indicates very few c-Fos positive cells in these mice. The authors need to show more representative images.

3. The authors report that decreasing or increasing the activity of Sim1 neurons in the PVH does not alter food intake. This is a surprising observation and is in conflict with the literature. For example, Sim1 neurons cover many groups of feeding-relevant PVH neurons, including MC4R neurons and prodynorphin neurons, and activation of these two groups of neurons has been shown to suppress food intake. The discrepancy could result from the small sample size used in the experiments. In Figure 4k, the variability in food intake is large, and 4 mice per group is not sufficient. The authors should do a power analysis to determine the sample size for the experiment.

4. Figure 3: There are many groups of PVH neurons that suppress appetite and promote energy expenditure. Why did the authors find that chronic activation of Sim1 neurons in the PVH decreased energy expenditure and caused obesity? As the authors found that deleting the $\gamma 2$ gene in the PVH leads to substantial apoptosis, they should determine if the phenotypes in NachBac-expressing mice result from cell death. Furthermore, it appears that NachBac-expressing mice had lower basal metabolism than control mice. Do the authors know the reason?

5. Supplementary Figure 4: A Nissl- or DAPI-stained image would better show the morphology of the PVH. It appears that there are too many cells positive to cleaved caspase-3 in mutant mice. Caspase-3 is only activated in dying cells, which may only last for a few hours. It is hard to image that there are so many dying cells at the moment when the authors kill the mice.

6. The value of y axis in many graphs does not start with 0, which could lead to a misleading impression of the data.

Reviewer #2 (Remarks to the Author):

In general, I find this work showing that BMAL1 is relevant for PVH neuronal activity to promote proper metabolic functions is of interest and value. All the results and data are straight forward and informative. My main concern regarding the publication of this work in Nature Communications are the following:

1 The PVH contains a diverse population of neurochemically and functionally different neurons, all of which may be Sim-1 expressing. Sim1 expression is not limited to the PVH. Thus mechanistically, the overall results are limited.

2) There is no control here to show that this rhythmicity of PVH neurons governed by BMAL1 and whether it is the only determinant of diurnal rhythms of metabolism.

Because of the above, I believe that the paper, as is, is better suited for a specialist journal.

Reviewer #3 (Remarks to the Author):

In this paper by Kim, Xu and colleagues, the authors report that deletion of BMAL1, a core clock gene and transcription factor, in neurons of the paraventricular hypothalamic (PVH) nucleus produced 1) a reduction in the diurnal rhythmicity in metabolism, and 2) an obese mouse. The authors moreover report that this same manipulation rendered these neurons incapable of responding to fast-refeeding. The reported findings are very interesting and suggest a heretofore unrecognized role for the molecular clock within neurons of the PVH in regulating diurnal rhythms of metabolism.

Major comments

I had to read this paper several times in order to fully understand what the authors did and, more importantly, why, i.e., understand the rationale. Hence the 'readability' of the paper is a major concern for me and I strongly feel that the flow and overall construction of the narrative needs to be reconsidered and polished (i.e., syntax of English, grammar).

Given also the high-visibility profile of the journal, I think a far more compelling case for publication could be made if the authors were to identify and functionally characterize at least one (presumptively hypothalamic) source of the GABAergic input to the PVH, in particular as the afferent 'arm' of this circuit seems to be central to understanding how and why these remarkable phenotypes would develop "in the real world" as well as for identifying potential therapeutic targets.

Minor comments

The inhibition of PVH neurons shown following expression of the Kir2.1 channels is compelling, but I am struggling to interpret the traces in Supp fig 2 panel D. In particular, the membrane potential appears to go to -150mV.

The results on expressing the NachBac channels are much more modest, if potentially misleading. For example, the effect on increasing neuronal firing appears to be driven by just 2 neurons that happen to have an unusual high frequency

While I may simply be confused, I am puzzled by the fact that both reducing PVH activity (Kir2.1 expression) and enhancing it produce the same phenotype of obesity. Can the authors please clarify?

Please include representative recordings of the sIPSCs and sEPSCs in Figs 4 and 5.

At what membrane potential were the IPSCs and EPSCs recorded?

Page 4, line 5: "TH" is not previously defined

Author rebuttal, first version:

Responses to reviewers' comments

We would like to thank all reviewers for their insightful comments. The following is our point-to-point responses to these comments.

Reviewer #1 (Remarks to the Author):

1. *Stereotaxic injection of AAV in Figure 1 and Figure 5 is not specific to the PVH, which makes it difficult to conclude that BMAL1 and $\gamma 2$ in the PVH regulate metabolic rhythms. For example, AAV-Cre-GFP infection is present in the subparaventricular zone and the periventricular hypothalamic nucleus in addition to the PVH (Fig. 1d). In fact, BMAL1 expression in these two structures is higher than in the PVH (Fig. 1b). Thus, the observed phenotypes could be a result of BMAL1 deletion in these two structures.*

Response: In this study, in order to delete BMAL1 and Y2 in the PVH of adult mice, we employed AAV-Cre-GFP delivery to the PVH. Currently, this is the only effective way to achieve gene deletion in adult mice. Inherent with this delivery, there will be some level of variation among study subjects. However, for this study, this will not cause a significant concern and will not alter the major conclusion drawn from this study. **A)** Due to the exact concern on variation with AAV-Cre-GFP injections, we didn't draw conclusion solely relying on data from these AAV-Cre-GFP injection experiments. On the contrary, we have spent significant efforts in using mouse models with PVH neurons clamped at a low (Kir2.1) and high (NachBac) with specific expression of Kir2.1 and NachBac in the PVH. These specific manipulations recapitulate the phenotypes of BMAL1 deletion, strongly support the role of PVH BMAL1 in the observed phenotype. **B)** In particular, since its expression is only selective in the PVH (Figs. 4a and 5b), $\gamma 2$ deletion by slight variation in AAV-Cre-GFP expression will be largely specific to the PVH. **C)** To specifically address the reviewer's concern on BMAL1 and to avoid potential confusion to readers, we have updated with a new case in which the AAV-Cre-GFP injection spared subparaventricular zone and periventricular PVH areas while exhibiting the same physiological phenotype in diurnal rhythms.

2. *Figures 1o-q: c-Fos immunoreactivity shown in the figures is not consistent with the quantification. The images show that there are many c-Fos positive cells in fasted mice and Bmal1 mutant mice, while the bar graph indicates very few c-Fos positive cells in these mice. The authors need to show more representative images.*

Response: We thank the reviewer for pointing out this "seemingly" inconsistency. The representative pictures (Figure 1o) show only one section of one brain while the bar graph (Figure 1q) shows the average of 3 sections of one brain. To avoid this confusion, we have update the Figure 1o with a more representative image.

3. *The authors report that decreasing or increasing the activity of Sim1 neurons in the PVH does not alter food intake. This is a surprising observation and is in conflict with the literature. For example, Sim1 neurons cover many groups of feeding-relevant PVH neurons, including MC4R neurons and prodynorphin neurons, and activation of these two groups of neurons has been shown to suppress food intake. The discrepancy could result from the small sample size used in the experiments. In Figure 4k, the variability in food intake is large, and 4 mice per group is not sufficient. The authors should do a power analysis to determine the sample size for the experiment.*

Response: We agree with the reviewer that Figure 4 on NachBac studies has issues with sample size. We have repeated this experiment and update all relevant data in Figure 4. The new body weight data recapitulate the previous body weight data, and therefore both sets of body weight were combined.

The reviewer also raised an issue on the phenotype on feeding versus energy expenditure in the PVH neurons. The major reason may lie in the time point when food intake is measured and compared. a) Previous studies reporting feeding effects were conducted by either acute changes in neuron activity (optogenetics or chemogenetics) (1, 2) or in mice with existing body weight difference (3, 4), and it is conceivable that the feeding difference is amplified. b) In many studies using animal models with PVH specific manipulations (MC4Rs, glutamate release etc), food intake is not obviously different when feeding is measured at a time point before body weight becomes obviously different (5, 6) (the body weight dependent effect of feeding is particularly evident in (6)); c) acute activation of MC4R neurons in transiently reducing feeding may not be necessarily translated into the prediction on chronic effects in reducing body weight as shown below on our preliminary data of NachBac expression in MC4R neurons in response to your Comment 4; and d) in the current study, we measured food intake with 2 weeks after viral delivery in CLAMS when the body weight is not that different. Therefore, our results on feeding is not conflicting with literature. For your convenience, I have attached Fig. 1 and Fig. 2 of Reference 6 below to demonstrate the data that the hyperphagia effect of MC4R knockout mice depends on the age and obesity of these mice.

Fig. 1 of Reference 6

Fig. 2 of Reference 6

One important point is that, as we focus on diurnal feeding patterns, we used CLAMS to monitor 24 hr energy expenditure and feeding continuously at an earlier time point before body weight diverges, which is to eliminate potential secondary effects from obesity. It is notable that our measurements from all animal models exhibited more pronounced disruption in energy expenditure than in feeding, i.e. more reduction in energy expenditure than in feeding during dark periods, suggesting a higher feeding

deficiency, which may drive the observed obesity development. Specifically for the NachBac mice, we measured CLAMS 3 weeks after viral delivery, in which the NachBac mice had a slightly higher body weight, which will not confound our data interpretation on diurnal rhythms, but may be part of reason for a slight decrease in baseline energy expenditure in data presented in Fig. 3f.

In order to clarify this issue to readers, we have added this point to Discussion as following:

“PVH neurons are known to regulate feeding as demonstrated by previous studies on prominent effects on feeding with optogenetic or chemogenetic manipulation of the activity of these neurons (40,46) or in obese animal models with altered PVH neuron function (42,47). It is intriguing that obesity in all animal models reported here was not associated with a significant change in feeding but rather with reduced energy expenditure. The major reason may be due to the fact that our CLAMS measurement was performed at an early time point, weeks 2 and 3 after viral delivery when there was no or litter body weight difference. Notably, earlier studies in animal models on PVH neurons reporting hyperphagia measured food intake at a time point with considerable obesity development (39,45) and it is conceivable that the feeding difference is amplified. In contrast, when food intake is measured before obesity development in animal models with PVH specific manipulations of MC4Rs or glutamate release, food intake is not obviously different (48,49). In particular, an obesity dependent effect of food intake difference has been nicely demonstrated in PVH neuron specific manipulation of MC4Rs4 (9). Given the tight association in diurnal patterns between feeding and energy expenditure, the PVH may regulate food intake and energy expenditure simultaneously. The animal models presented here all exhibited a more pronounced disruption in diurnal patterns in energy expenditure than in feeding, specifically, more reduction in energy expenditure than feeding during dark periods, suggesting a higher feeding efficiency, which may drive obesity development.”

4. Figure 3: There are many groups of PVH neurons that suppress appetite and promote energy expenditure. Why did the authors find that chronic activation of Sim1 neurons in the PVH decreased energy expenditure and caused obesity? As the authors found that deleting the $\gamma 2$ gene in the PVH leads to substantial apoptosis, they should determine if the phenotypes in NachBac-expressing mice result from cell death. Furthermore, it appears that NachBac-expressing mice had lower basal metabolism than control mice. Do the authors know the reason?

Response: As discussed before, the effect of PVH neurons to suppress feeding (i.e. activation of PVH neurons reduces feeding) is largely acute (i.e. short term effects on feeding with acute activation) (1, 2). How these effects can be translate to chronic effects such as daily feeding and obesity is unknown. The effect on energy expenditure seems to be both acute and chronic, as clearly demonstrated by our data when it was measured in earlier (2 weeks after viral delivery) or later (8 weeks after viral delivery), as shown in Fig. 1. As the reviewer mentioned, the PVH consists of many diverse groups of neurons and interestingly. While some groups of PHV neurons acutely inhibit feeding (1, 2), TRH neurons show a surprising potent and acute food intake-increasing effect (7). In addition, the effect of TRH neurons and a few other neurons on energy expenditure is unknown. So it cannot be easily predictable on the overall effect of chronic PVH neuron activation based on literature.

Although it is easy to predict that reducing PVH activity (Kir2.1) causes obesity as most studies demonstrate this point. It is indeed somewhat counter-intuitive to observe that chronic activation of PVH neurons also lead to obesity. As mentioned above, careful literature reading finds that most studies on the activation of PVH neurons (or subset of those) in reducing feeding have been conducted in an acute

fashion (i.e. feeding effects in a range of a few hours). However, despite the known implication of PVH neurons in feeding inhibition, an animal model on PVH neurons with a reduced body weight phenotype is yet to be reported. On the contrary, in most chronic animal models with “presumptive effects on PVH neuron activation”, including arcuate POMC neuron activation, NPY deletion in adult mice, PVH CRH neuron activation (deletion of glucocorticoid receptor GR), all lead to no obvious

reduction in body weight on chow diet (5, 6, 8-10), and , in opposite, GR deletion causes slight body weight increase (11).

Our very preliminary data showed that, while chronic inhibition of PVH MC4R neurons (expression of Kir2.1 using MC4R-Cre, panels A and C in Figure shown below) causes obesity, chronic activation of PVH MC4R neurons (expression of NachBac using MC4R-Cre, panels B and C in Figure shown below) show no obvious change in body weight on chow diet. Thus, it appears that, although reducing PVH neuron activity causes obesity, activating it may not necessarily causes an opposite effect in body weight.

Our data showed chronic activation of PVH neurons will reduce their responses to endogenous signals as demonstrated in our *in vivo* fiber photometric data (Fig. 7), which may reduce PVH neuron “sensitivity” in response of energy expenditure and feeding dynamics to internal and external environmental changes. This alteration may manifest in behavior as disrupted diurnal rhythms in energy expenditure and feeding. Since our animal models exhibited a more pronounced reduction in energy expenditure than in feeding during dark periods, suggesting a higher feeding efficiency, driving obesity.

We have provided Caspase 3 staining in brain section of NachBac mice (right), which showed no difference cleaved caspase 3 expression (blue) between the 2 groups, arguing against the possibility of loss of neurons by NachBac. The control group is represented by top 2 panels and the NachBac group is represented by the bottom 3 panels. We used Sim1-Cre Ai9 reporter mice to show tissue morphology. This set data has been added as new Supplementary Fig. 4. In addition, our preliminary experiments with divergent effects of PVH MC4R-expressing neurons with NachBac and Kir2.1 also argue against the possibility of NachBac expressing causing a loss of neuron effect.

To address this important concern and to avoid confusion, we have added the following discussion.

“One potential explanation may lie in the difference between acute versus chronic alterations in PVH neuron activity. Despite previous studies implicating acute activation of various subsets of PVH neurons in feeding inhibition (2, 12), an animal model on these neurons presented with a lean body weight phenotype, the presumptive phenotype from reduced feeding, has yet to be reported. Although further studies are warranted, this phenomenon may indicate that, contrary to the prediction from PVH neuron inhibition in obesity development, activation of these neurons

is not sufficient to reduce body weight. Supporting this, while inhibition of proopiomelanocortin (POMC) neurons, one key upstream neurons of PVH melanocortin receptor 4 (MC4R) neurons, causes massive obesity, activation of these neurons fails to produce a lean phenotype (9)".

5. *Supplementary Figure 4: A Nissl- or DAPI-stained image would better show the morphology of the PVH. It appears that there are too many cells positive to cleaved caspase-3 in mutant mice. Caspase-3 is only activated in dying cells, which may only last for a few hours. It is hard to image that there are so many dying cells at the moment when the authors kill the mice.*

Response: We have provided additional pictures for caspase 3 activation. We used NeuN, a neuron marker, instead to show the morphology. We have updated this supplementary Figure to reflect the updated data.

We agree with Caspase 3 activation in dying cells, but not sure on how long it lasts in neurons. We did a brief search and found information from the course: https://beta-static.fishersci.com/content/dam/fishersci/en_US/documents/programs/scientific/brochures-and-catalogs/publications/promega-timing-apoptosis-assay-publication.pdf, in which it described that Caspase 3 can be in an activated state with a peak not earlier than 7 hours. Published studies also showed that Caspase 3 may remain to be detected as long as 25 or 72 hours (13, 14). For your convenience, I have attached one figure showing detectable Caspase 3 72 hours after treatment below.

It is conceivable that the period for us to be able to detect activated caspase 3 in various number of neurons is quite long. In addition, other studies with a similar condition also reported a similar level of Caspase 3 activation (15, 16). Therefore, the amount of Caspase 3 we observed is not unreasonable.

6. *The value of y axis in many graphs does not start with 0, which could lead to a misleading impression of the data.*

Response: We have updated the y axis for body weight with a start point at 0. However, for energy expenditure, if we make a change, then we will not be able to appreciate the difference given the nature of change in small percentage relative to baseline, which will defeat the purpose of figure presentation. Thus, we have maintained the way of presentation for energy expenditure.

Reviewer #2 (Remarks to the Author):

1 The PVH contains a diverse population of neurochemicaly and functionally different neurons, all of which may be Sim-1 expressing. Sim1 expression is not limited to the PVH. Thus mechanistically, the overall results are limited.

Response: We appreciate the reviewer's point on the PVH with diverse group of neurons; however, we respectfully disagree with the reviewer's assessment that our overall results are limited. The prevalent studies on feeding focus on static regulation of feeding and energy expenditure. As diet-induced obesity, the major contributing factor to the current obesity epidemic, is known to be associated with defective diurnal patterns in feeding and energy expenditure, the brain mechanism underlying diurnal patterns remains unclear. The current study represents the very first study to identify the PVH as one major brain site orchestrating diurnal metabolism. Importantly, we also demonstrated that HFD disrupted PVH neuron responsiveness, which points to a novel fundamental metabolism underlying diet-induced obesity (DIO) at a cellular level. Therefore, the current study bridges circadian regulation to PVH function, identifies neuron responsiveness as a key cellular mechanism underlying circadian regulation and DIO, and is distinct from other studies adding another piece of evidence for PVH neurons in body weight regulation.

Our studies aim to delete BMAL1 and $\gamma 2$ in the PVH of adult mice. Currently, it is technically challenging to achieve this in specific subsets of PVH neurons. We are currently pursuing new ways to achieve this. However, as discussed above, the focus on this study is to investigate the novel role of the PVH in diurnal regulation of metabolism, and the study on the contribution of specific subsets of PVH neurons to this novel function represents a natural extension of this study. Thus, publication of this study will be informative and allow peers in this field to come up with a suitable approach to interrogate the function of individual groups of PVH neurons.

The reviewer's concern on "Sim1 expression is not limited to the PVH" is probably caused by a misunderstanding. Since we use Sim1-Cre and deliver viral vectors specific to the PVH. Those non-PVH Sim1 expressing brain areas are not targeted in the current study.

2) There is no control here to show that this rhythmicity of PVH neurons governed by BMAL1 and whether it is the only determinant of diurnal rhythms of metabolism.

Response: For the comment "*There is no control here to show that this rhythmicity of PVH neurons governed by BMAL1*", we assume that the reviewer suggest that there is no control showing that the rhythmicity of PVH neuron activity is governed by BMAL1. We agree with the reviewer on this point and it would be ideal to show that deletion of BMAL1 results in loss of diurnal activity pattern of the PVH. However, it is currently technical challenging for us to achieve direct measurement of PVH neuron activity across daily 24 hrs. To overcome this shortcoming, we have gone extreme to use additional ways, including clamping PVH neurons at a high (NachBac) and a low (Kir2.1), and clamping GABA-A receptor $\gamma 2$ subunit expression. In particular, our data show that $\gamma 2$ exhibits a diurnal pattern of expression and BMAL1 controls $\gamma 2$ expression, which is known to be a potent contributor to inhibitory GABAergic input. Consistently, we also demonstrate that PVH neurons exhibit diurnal patterns in IPSCs, a major determinant of neuron activity and that the diurnal pattern is disrupted by clamped $\gamma 2$ expression. These results taken together strongly support that BMAL1 controls PVH neuron diurnal activity pattern. However, it is also important to point out that, due to the exact reason that it is technically difficult for us

to directly demonstrate PVH neuron diurnal activity pattern, we refrain from making claims on diurnal activity of PVH neurons in the manuscript, but instead focusing on neuron responsiveness, for which we have compelling evidence. In the manuscript, our over-arching conclusion is BMAL1-^{*} γ 2-^{*} responsiveness -^{*} diurnal rhythms, and only provided a discussion point to speculate that disruption of PVH neuron responsiveness may lead to alterations of diurnal PVH neuron activity.

Thus, even without data on diurnal PVH neuron activity pattern, it will not affect the major conclusions drawn from the data presented.

For the comment on “*whether it is the only determinant of diurnal rhythms of metabolism*”, our data show that it is not the only determinant as our animal models on GABA-A receptor $\gamma 2$ subunit also show a similar phenotype. It is conceivable that there might be BMAL1-dependent and -independent pathways that may be involved in controlling PVH neuron responsiveness and diurnal rhythms. However, in any case, our data convincingly demonstrate that the BMAL1 pathway is required for diurnal rhythms.

Reviewer #3 (Remarks to the Author):

Major comments

1. I had to read this paper several times in order to fully understand what the authors did and, more importantly, why, i.e., understand the rationale. Hence the ‘readability’ of the paper is a major concern for me and I strongly feel that the flow and overall construction of the narrative needs to be reconsidered and polished (i.e., syntax of English, grammar).

Response: We are sorry and apologize for our poor readability. This may in part be due to the amount of data from 5 different animal models presented in this study. As illustrated in our revised version of the manuscript, we have substantially edited and added necessary discussion points to improve the narrative for a better logic follow.

2. Given also the high-visibility profile of the journal, I think a far more compelling case for publication could be made if the authors were to identify and functionally characterize at least one (presumptively hypothalamic) source of the GABAergic input to the PVH, in particular as the afferent ‘arm’ of this circuit seems to be central to understanding how and why these remarkable phenotypes would develop “in the real world” as well as for identifying potential therapeutic targets.

Response: We appreciate the reviewer’s comment on the afferent “arm”, which is important to appreciate the underlying physiology associated with the importance of this study. As a matter of fact, the current study has been “directed” by one of previous studies on the function of GABAergic input to

the PVH (17). For your convenience, I've illustrated the main results from the study below. Basically, we used a mouse model of Pdx1-Cre::Vgatflox/flox mice. Pdx1-Cre expresses Cre mainly in the Arc and lateral hypothalamic neurons and these Cre-expressing neurons project extensively to the PVH. Vgat (vesicular GABA transporter) is required for presynaptic GABA release. In this mice, we found that GABA release to the PVH is substantially reduced (top panel of the presented in the above, (17)) and these mice exhibited defective diurnal patterns of feeding and energy expenditure (bottom panel, (17)). Additional supporting evidence in this regard is that AgRP

neurons are known to send GABAergic input to the PVH and exhibit diurnal pattern of activity (18). These results suggest that GABAergic neurons in the arcuate and lateral hypothalamus, including AgRP neurons, are part of the afferent arm to the PVH in orchestrating diurnal metabolism. To make this point more clear, we have added this sentence in our existing discussion on presynaptic GABAergic inputs “One of major sources of GABAergic inputs may be from the Arc and lateral hypothalamus as our previous results demonstrate disruption of GABA release from these sites substantially reduces GABAergic input to PVH neurons and blunted diurnal rhythms in feeding and energy expenditure (17).”

Minor comments

1. The inhibition of PVH neurons shown following expression of the Kir2.1 channels is compelling, but I am struggling to interpret the traces in Supp fig 2 panel D. In particular, the membrane potential appears to go to -150mV.

Response: To explain this unusual phenomenon, we may consider the following: (1) theoretically, pure ion channel activities should not bring neuron’s membrane potential to below -100 mV under normal physiological conditions; (2) If the membrane potential crosses an ion channel’s reversal potential, ions will flow to the opposite direction. For potassium channels, including Kir2.1, if we assume that the reversal potential is around -90 mV, hyperpolarizing to more negative than -90 mV will cause K⁺ to flow from outside into inside of the neuron, which will depolarize the membrane potential. As we injected negative currents, which means negative charges were injected into the cytoplasm, leading to a low -150 mV membrane potential. On the other hand, during negative current injection, ion flow across the plasma membrane in control neurons was limited, thus the input resistance was high in control neurons. By contrast, during negative current injection, the injected negative charges were neutralized by the K⁺ influx evoked by the opening of Kir2.1 channels when the membrane potential was more hyperpolarized than -90 mV in Kir2.1 expressing neurons. As a result, the membrane potential hyperpolarization in those neurons was much less than that in control neurons, leading to much lower calculated input resistance in Kir2.1 expressing neurons, which reflects a significant plasma membrane conductance increase due to the activities of overexpressed Kir2.1 channels.

2. The results on expressing the NachBac channels are much more modest, if potentially misleading. For example, the effect on increasing neuronal firing appears to be driven by just 2 neurons that happen to have an unusual high frequency

Response: One key factor to consider that the NachBac currents are much slower in inactivation and have a much lower threshold in firing. Therefore, each firing in NachBac-expressing neurons will bring in much more cation to the cell, i.e. the efficiency of each firing in NachBac neurons in increasing Ca²⁺ is much greater than the control neurons. Thus, even with a comparable number of action potential firing, NachBac neurons will have a high level of excitability. Consistently, the vast majority, if not all, of these neurons exhibited c-Fos, a marker for neuron excitation (Fig. 3a).

3. While I may simply be confused, I am puzzled by the fact that both reducing PVH activity (Kir2.1 expression) and enhancing it produce the same phenotype of obesity. Can the authors please clarify?

Response: We appreciate the reviewer in raising this important point. Both conditions (increasing or reducing activity) cause obesity with reduced diurnal metabolism powerfully demonstrate the point that normal diurnal metabolism is an important factor for normal body weight regulation. While it is easy to

predict that reducing PVH activity (Kir2.1) causes obesity as most studies demonstrate this point, it is indeed counter-intuitive to observe that chronic activation of PVH neurons also lead to obesity. However, it is worth noting that the effect of PVH neurons to suppress feeding (i.e. activation of PVH neurons reduces feeding) is largely acute (i.e. short term effects on feeding with acute activation) (1, 2). How these effects can be translated to chronic effects such as daily feeding and obesity is unknown. While some groups of PVH neurons acutely inhibit feeding (1, 2), TRH neurons show a surprising potent and acute food intake-increasing effect (7). In addition, the effect of TRH neurons and a few other neurons on energy expenditure is unknown. Our very preliminary data showed that, while chronic inhibition of PVH MC4R neurons causes obesity, chronic activation of PVH MC4R neurons (expression of NachBac in PVH MC4R-Cre expressing neurons) show no obvious change in body weight on chow diet (Please see the MC4R figure shown in the response to Reviewer 1). Thus, it appears that, although reducing PVH neuron activity causes obesity, activating it may not necessary causes an opposite effect in body weight change. So it cannot be easily predictable on the overall effect of chronic PVH neuron activation.

Specifically on feeding inhibitory effects by PVH neuron activation, careful literature reading finds that most studies on the activation of PVH neurons (or subset of those) in reducing feeding have been conducted in an acute fashion (i.e. feeding effects in a range of a few hours) (1, 2). However, despite the known implication of PVH neurons in feeding inhibition, an animal model on PVH neurons with a reduced body weight phenotype is yet to be reported. On contrary, in most chronic models with “suspected effects in PVH neuron activation”, including arcuate POMC neuron activation, NPY deletion in adult mice, PVH CRH neuron activation (deletion of corticosterone receptor GR), all lead to no obvious reduction in body weight on chow diet (5, 6, 8-10), and, in opposite, GR deletion causes slight body weight increase (11).

As discussed previously in response to Reviewer 1, our animal models exhibited a more pronounced reduction in energy expenditure than in feeding during dark periods, suggesting a higher feeding efficiency, which is sufficient to drive obesity development. Our collective results demonstrates the importance of diurnal metabolism to body weight regulation. We have provided explanation in the manuscript in the Discussion, as shown in our response to Reviewer 1.

4. Please include representative recordings of the sIPSCs and sEPSCs in Figs 4 and 5.

Response: We have included both recordings in new Supplementary Fig. 5.

5. At what membrane potential were the IPSCs and EPSCs recorded?

Response: We have included this information in the method (-40mV).

6. Page 4, line 5: “TH” is not previously defined

Response: We have corrected it.

References:

1. M. M. Li *et al.*, The Paraventricular Hypothalamus Regulates Satiety and Prevents Obesity via Two Genetically Distinct Circuits. *Neuron* **102**, 653-667 e656 (2019).
2. A. K. Sutton *et al.*, Control of food intake and energy expenditure by Nos1 neurons of the paraventricular hypothalamus. *J Neurosci* **34**, 15306-15318 (2014).
3. N. Balthasar *et al.*, Divergence of melanocortin pathways in the control of food intake and energy expenditure. *Cell* **123**, 493-505 (2005).
4. D. Xi, N. Gandhi, M. Lai, B. M. Kublaoui, Ablation of Sim1 neurons causes obesity through hyperphagia and reduced energy expenditure. *PloS one* **7**, e36453 (2012).
5. Y. Xu *et al.*, Glutamate mediates the function of melanocortin receptor 4 on Sim1 neurons in body weight regulation. *Cell Metab* **18**, 860-870 (2013).
6. K. R. Vella *et al.*, NPY and MC4R signaling regulate thyroid hormone levels during fasting through both central and peripheral pathways. *Cell Metab* **14**, 780-790 (2011).
7. M. J. Krashes *et al.*, An excitatory paraventricular nucleus to AgRP neuron circuit that drives hunger. *Nature* **507**, 238-242 (2014).
8. L. M. Andino *et al.*, POMC overexpression in the ventral tegmental area ameliorates dietary obesity. *J Endocrinol* **210**, 199-207 (2011).
9. C. Zhan *et al.*, Acute and long-term suppression of feeding behavior by POMC neurons in the brainstem and hypothalamus, respectively. *The Journal of neuroscience : the official journal of the Society for Neuroscience* **33**, 3624-3632 (2013).
10. L. Ste Marie, S. Luquet, T. B. Cole, R. D. Palmiter, Modulation of neuropeptide Y expression in adult mice does not affect feeding. *Proceedings of the National Academy of Sciences of the United States of America* **102**, 18632-18637 (2005).
11. G. Laryea, G. Schutz, L. J. Muglia, Disrupting hypothalamic glucocorticoid receptors causes HPA axis hyperactivity and excess adiposity. *Mol Endocrinol* **27**, 1655-1665 (2013).
12. A. S. Garfield *et al.*, A neural basis for melanocortin-4 receptor-regulated appetite. *Nature neuroscience* **18**, 863-871 (2015).
13. R. K. Kasam, G. B. Reddy, A. G. Jegga, S. K. Madala, Dysregulation of Mesenchymal Cell Survival Pathways in Severe Fibrotic Lung Disease: The Effect of Nintedanib Therapy. *Front Pharmacol* **10**, 532 (2019).
14. P. Wongchitrat *et al.*, Elevation of Cleaved p18 Bax Levels Associated with the Kinetics of Neuronal Cell Death during Japanese Encephalitis Virus Infection. *Int J Mol Sci* **20**, (2019).
15. A. Gunther, V. Luczak, T. Abel, A. Baumann, Caspase-3 and GFAP as early markers for apoptosis and astrogliosis in shRNA-induced hippocampal cytotoxicity. *J Exp Biol* **220**, 1400-1404 (2017).
16. W. C. Huang, R. Abraham, B. S. Shim, H. Choe, D. T. Page, Zika virus infection during the period of maximal brain growth causes microcephaly and corticospinal neuron apoptosis in wild type mice. *Sci Rep* **6**, 34793 (2016).
17. E. R. Kim *et al.*, Hypothalamic Non-AgRP, Non-POMC GABAergic Neurons Are Required for Postweaning Feeding and NPY Hyperphagia. *The Journal of neuroscience : the official journal of the Society for Neuroscience* **35**, 10440-10450 (2015).

18. T. Liu *et al.*, Fasting activation of AgRP neurons requires NMDA receptors and involves spinogenesis and increased excitatory tone. *Neuron* **73**, 511-522 (2012).

Reviewer comments, second version:

Reviewer #1 (Remarks to the Author):

The authors have addressed many of my previous concerns in this resubmission; however, some of my concerns remain.

1. c-Fos immunohistochemistry in Fig. 1o and p does not appear to be specific. For example, there are many c-Fos-positive cells in the image for fasting control. That is the reason why I raised the concern that the images and quantification are not consistent. It is not clear how the authors distinguished between real positives and non-specific staining. To authors' credit, the representative c-Fos images in Figures 2 and 3 look much better. It is necessary to fix the high c-Fos background issue in Figure 1 before the paper is accepted for publication.

2. Supplementary figure 4 does not address the concern that neuronal over-excitation by NachBac expression could kill PVH neurons. This is a serious concern because the reduction in energy expenditure in NachBac-expressing mice could result from death of some PVH neurons. Supplementary figure 4 only shows apoptotic cells at a single time point. If susceptible neurons had died and been removed by glial cells before the tissue samples were taken, the authors would not detect any apoptotic cells. The authors could perform Sim1 immunohistochemistry and count positive neurons in the PVH several weeks after AAV injection to address the concern.

Reviewer #2 (Remarks to the Author):

The authors adequately addressed my comments. The paper is stronger with the adjustments they did to all of the comments of the reviewers.

Reviewer #3 (Remarks to the Author):

The authors have addressed my previously articulated concerns and points of confusion. I congratulate them on a nicely executed and timely study.

Author rebuttal, second version:**Responses to reviewers' comments**

We would like to thank all reviewers for their insightful comments. The following is our point-to-point responses to these comments.

Reviewer #1 (Remarks to the Author):

The authors have addressed many of my previous concerns in this resubmission; however, some of my concerns remain.

1. c-Fos immunohistochemistry in Fig. 1o and p does not appear to be specific. For example, there are many c-Fos-positive cells in the image for fasting control. That is the reason why I raised the concern that the images and quantification are not consistent. It is not clear how the authors distinguished between real positives and non-specific staining. To authors' credit, the representative c-Fos images in Figures 2 and 3 look much better. It is necessary to fix the high c-Fos background issue in Figure 1 before the paper is accepted for publication.

Response: We thank the reviewer for raising the issue on the c-Fos immunostaining in fasting control. The c-Fos immunostaining in red (this case) has a tendency to have a higher level of background fluorescence, compared to the green ones (Figs. 2 and 3). We count only those neurons with round (nucleus like) and bright red staining as positive. We have updated the fasting control panel with a more representative image to prevent potential confusion from readers.

2. Supplementary figure 4 does not address the concern that neuronal over-excitation by NachBac expression could kill PVH neurons. This is a serious concern because the reduction in energy expenditure in NachBac-expressing mice could result from death of some PVH neurons. Supplementary figure 4 only shows apoptotic cells at a single time point. If susceptible neurons had died and been removed by glial cells before the tissue samples were taken, the authors would not detect any apoptotic cells. The authors could perform Sim1 immunohistochemistry and count positive neurons in the PVH several weeks after AAV injection to address the concern.

Response: We agree that our Supplementary Figure 4 only shows whether cells undergo apoptosis at a single time point. However, if NachBac expression causes neuronal over-excitation and death, it is most unlikely that the expression will only selectively cause over-excitation and complete death on one subset of neurons while having no effects on the remaining subset of neurons. In any case, as suggested by the reviewer, we have provided additional evidence shown below to demonstrate the Sim1 neuron number had no alterations between groups. As the control mice are Sim1-Cre: Ai9 reporter mice with injection of AAV-Flex-GFP, we used Reporter (red) instead to show Sim1 neurons (Top panels). In NachBac injected mice, we used Sim1 immunostaining to show Sim1 neurons (Red) and NachBac expression was shown in green (bottom panels). Quantification of Sim1 neuron number shows no difference in number between groups, suggesting no neuronal death.

Reviewer comments, third version:

REVIEWERS' COMMENTS:
Reviewer #1 (Remarks to Author):

The authors have addressed the remaining concerns. I think that the manuscript is ready for publication.

Author rebuttal, third version: